# Expected Improvement for Contextual Bandits

**Hung Tran-The**[*]
Applied Artificial Intelligence Institute
Deakin University

**Sunil Gupta**
Applied Artificial Intelligence Institute
Deakin University

**Santu Sana**
Applied Artificial Intelligence Institute
Deakin University

**Tuan Truong**
FPT Software AI Center

**Long Tran-Thanh**
University of Warwick, UK

**Svetha Venkatesh**
Applied Artificial Intelligence Institute
Deakin University

## Abstract

The expected improvement (EI) is a popular technique to handle the tradeoff between exploration and exploitation under uncertainty. This technique has been widely used in Bayesian optimization but it is not applicable for the contextual bandit problem which is a generalization of the standard bandit and Bayesian optimization. In this paper, we initiate and study the EI technique for contextual bandits from both theoretical and practical perspectives. We propose two novel EI based algorithms, one when the reward function is assumed to be linear and the other for more general reward functions. With linear reward functions, we demonstrate that our algorithm achieves a near-optimal regret. Notably, our regret improves that of LinTS [3] by a factor $\sqrt{d}$ while avoiding to solve a NP-hard problem at each iteration as in LinUCB [1]. For more general reward functions which are modeled by deep neural networks, we prove that our algorithm achieves a $\tilde{\mathcal{O}}(\tilde{d}\sqrt{T})$ regret, where $\tilde{d}$ is the effective dimension of a neural tangent kernel (NTK) matrix, and $T$ is the number of iterations. Our experiments on various benchmark datasets show that both proposed algorithms work well and consistently outperform existing approaches, especially in high dimensions.

## 1   Introduction

The contextual bandit problem is an important field in machine learning ([9, 23]) to optimize the trade-off between exploration and exploitation in sequential decision making, and has been extensively studied in real-world applications such as personalized recommendation [24], advertising [18], robotic control [25], and healthcare [19]. At each round, the agent observes a feature vector (the "context") for each of the $K$ arms, pulls one of them, and in return receives a scalar reward. The goal is to maximize the cumulative reward, or minimize regret (see our definition in Section 2), in a total of $T$ rounds. To do so, the agent must find a trade-off between exploration and exploitation.

There are two standard techniques to solve the trade-off in contextual bandits. The first technique uses the optimism in face of uncertainty which chooses promising actions by maximizing upper-confidence bounds (UCB), and the second one using Thompson Sampling (TS) whose basic idea is to estimate a posterior distribution on the reward, and sample an arm that maximises a random reward drawn

---

[*]Correspondence to Hung Tran-The `<hung.tranthe@deakin.edu.au>`.

36th Conference on Neural Information Processing Systems (NeurIPS 2022).

from this distribution. A series of work has applied both UCB and TS techniques or their variants to explore in contextual bandits with many forms of reward functions - from linear to nonlinear. In the line of UCB, there are works of [24, 13, 1] for the linear bandits, works of [17, 35] for nonlinear contextual bandits, and very recently [39], which uses neural networks to learn the reward function. In the line of TS, [3, 31] are for linear bandits, [31, 22] for generalized linear functions, and [30, 38] for nonlinear bandits using deep neural networks.

Different from UCB and TS, the expected improvement (EI) [26] is a greedy improvement-based heuristic that samples an action offering the greatest expected improvement over an incumbent. EI is also one of the oldest and popular techniques to handle the tradeoff between exploration and exploitation under uncertainty. Due to its ability to handle uncertainty, EI has been widely used in Bayesian optimization [28, 37] - a problem that is closely related to infinite-arm multi-arm bandits. However, to the best of our knowledge, EI has not been used in contextual bandits.

In addition, in many situations where the exploration may costly or infeasible, EI can be a safer strategy than UCB and TS which are considered as over-exploration strategies. For example, in a medical application of contextual bandits, choosing a treatment that is not the estimated-best choice (pure exploration) for a specific patient may be unethical [7]. Compared to UCB and TS strategies, EI brings a safer treatment because EI only chooses a treatment with high possibilities for improvement over the estimated-best choice.

Motivated by these advantages, it is valuable to study EI technique in the contextual bandit setting. Whether EI can handle well the trade-off between exploration and exploitation in a contextual bandit setting and further in reinforcement learning is an interesting question. Our main contributions in this paper are as follows:

- We introduce and formalize Expected Improvement as a new strategy for contextual bandits creating a parallel to UCB and TS.

- We propose two EI-based algorithms. The first algorithm (LinEI) is designed for the linear bandits whilst the second algorithm (NeuralEI) is designed for more general reward function and we model it by a deep neural network.

- For the linear reward function, our LinEI algorithm is able to achieve $\tilde{\mathcal{O}}(d\sqrt{T})$ regret with probability $1 - \delta$, which matches the information theoretic lower bound $\Omega(d\sqrt{T})$ for this problem (up to $\ln(T)$). Our regret improves that of LinTS [3] by a factor $\sqrt{d}$ while avoiding to solve a NP-hard problem at each iteration as in LinUCB [1]. By this advantage, we will show in section 6.1.1 that our proposed LinEI scales to high dimensions (in term of $d$) better than LinUCB and LinTS; in terms of computations, LinEI is also significantly cheaper than both LinUCB and LinTS. Our source code is publicly available at `https://github.com/Tran-TheHung/Expected-Improvement-for-Contextual-Bandits`.

- For the general reward function, we prove that, under standard assumptions (see section 4.1), our NeuralEI algorithm is able to achieve $\tilde{\mathcal{O}}(\tilde{d}\sqrt{T})$ regret, where $\tilde{d}$ is the effective dimension of a neural tangent kernel matrix and $T$ is the number of rounds. The regret bound of NeuralEI has the same order as those of NeuralUCB [39] and NeuralTS [38].

- Finally, we demonstrate the performance of our proposed EI-based algorithms against UCB and TS based approaches and other baselines on various benchmark datasets. Our experiments show that LinEI outperforms other baselines in the linear setting, and when the reward function is non-linear, NeuralEI outperforms all baselines.

Under a theoretical perspective, we remark that the EI technique have not been well studied compared to UCB and TS even in the Bayesian optimization setting. A key challenge of analyzing EI algorithms theoretically comes from its improvement function involving nonlinear, nonconvex term. A notable exception is the work in [11], which provides a convergence analysis of EI for Bayesian optimization in the noise-free setting. Another work in [34] provides the convergence analysis of EI for Bayesian optimization in the noisy setting. There are also several papers [33, 29] using EI to study the best-arm identification problem (also known as "pure exploration") which is a finite variant of Bayesian optimization. However, none of these works provides the optimal convergence rate in their settings. In contrast, our work is the first to show that EI can achieve the optimal convergence rate at least in linear contextual bandit setting.

## 2 Problem Setting

We consider the problem of $K$-arm contextual bandits. At time $t = 1, 2, ...$, the agent observes $K$ contextual vectors $x_{i,t} \in \mathbb{R}^d$, then selects an arm $a(t)$ and receives a reward $r_{a(t),t}$ which has a general form as follows:

$$r_{a(t),t} = h(x_{a(t),t}) + \xi_{a(t),t},$$

where $h$ is an unknown reward mean function satisfying $0 \leq h(x) \leq 1$ for any $x \in \mathbb{R}^d$, and $\xi_{a(t),t}$ is a zero mean, conditionally $R$-subGaussian noise with a constant $R \geq 0$, i.e., $\forall \lambda \in \mathbb{R}, \mathbb{E}[e^{\lambda \xi_{a(t),t}} | \{x_{i,t}\}_{i=1}^K] \leq \exp(\frac{\lambda^2 R^2}{2})$. In our setting, we assume that these context vectors may be chosen by an adversary in an adaptive manner after observing the arms played and their rewards up to time $t - 1$. For the unknown function $h$, we consider two cases as follows:

- The reward function $h$ is linear, i.e., $h(x_{t,i}) = x_{t,i}^T \theta^*$, where $\theta^* \in \mathbb{R}^d$ is fixed but unknown parameters. Without loss of generality, we here assume that $||x_{i,t}|| \leq 1, ||\theta^*|| \leq 1$.

- The reward function $h$ is modelled by a fully connected neural network with depth $L \geq 2$ defined recursively by

$$f(x; \theta) = \sqrt{m} W_L \sigma(W_{L-1} \sigma(...\sigma(W_1 x))),$$

where $\sigma(x) := \max\{x, 0\}$ is the ReLU activation, $\theta = (\text{vec}(W_1); ...; \text{vec}(W_L)) \in \mathbb{R}^p$ is the collection of parameters of the neural network with $p = dm + m^2(L - 2) + m$ which is the number of parameters of the network. Without loss of generality, we assume that the width of each hidden layer is the same (i.e., $m$) for convenience in analysis. We denote the gradient of the neural network function by $g(x; \theta) = \bigtriangledown_\theta f(x; \theta) \in \mathbb{R}^p$.

**Performance Measure.** Let $a^*(t)$ denote the optimal arm at time $t$. The objective is to minimize the cumulative regret $R(T) = \sum_{t=1}^T (x_{a^*(t),t}^T \theta^* - x_{a(t),t}^T \theta^*)$.

## 3 The LinEI algorithm for linear bandits

In this section, we design a provable version of EI algorithm for linear bandits. To do this, we make some assumptions on the prior distribution of the reward function before defining a form of the expected improvement.

**Prior and Posterior Distributions.** Inspired from the design of priors of the reward function like TS algorithm [3], we assume that the reward $r_{i,t}$ of each arm $i$ follows a Gaussian distribution $\mathcal{N}(x_{i,t}^\top \theta^*, v^2)$, where the variance $v^2$ is a free parameter (possibly time-dependent) that can be set specific to an algorithm. Let

$$X(t) = \lambda I + \sum_{j=1}^{t-1} x_{a(j),j} x_{a(j),j}^\top$$

$$\hat{\theta}_t = X(t)^{-1} (\sum_{j=1}^{t-1} x_{a(j),j} r_{a(j),j}).$$

Then if we assume that the prior for $\theta^*$ at time $t$ is given by $\mathcal{N}(\hat{\theta}_t, v^2 X^{-1}(t))$, then the posterior distribution of $\theta^*$ at time $t + 1$ is $\mathcal{N}(\hat{\theta}_{t+1}, v^2 X^{-1}(t+1))$ ( see the proof in Appendix A.1 in [3]).

**Expected Improvement.** We now use this posterior distribution update to define the form of the expected improvement of each arm in contextual bandits. We denote $r_t^+ = \max_{i \in \mathcal{K}}\{x_{i,t}^T \hat{\theta}_t\}$ which is the largest mean estimate of reward among all arms at time $t$. We define the expected improvement of an arm $i$ at time $t$ as

$$\alpha_{i,t}^{EI} = \mathbb{E}_{\mu \sim \mathcal{N}(\hat{\theta}_t, v_t^2 X(t)^{-1})}[\max\{0, x_{i,t}^\top \mu - r_t^+\}], \quad (1)$$

This form is similar to those of EI in the case of the multi-arm bandit [29] and in the case of Bayesian optimization [34], but for contextual bandits. The $\alpha_{i,t}^{EI}$ value measures the potential of arm $i$ to

---

**Algorithm 1** The Linear Expected Improvement Algorithm (LinEI)

---

**Input**: Parameters $C_0, \beta$

1: **for** $t = 1$ to $T$ **do**
2:     Observe contexts $\{x_{i,t}\}_{i=1}^K$
3:     Set $\bar{a}(t) := \operatorname{argmax}_{i \in \mathcal{K}} \alpha_{i,t}^{EI}$, $\tilde{a}(t) = \operatorname{argmax}_{i \in \mathcal{K}} \{x_{i,t}^\top \hat{\theta}_t\}$
4:     **if** $\alpha_{\bar{a}(t),t}^{EI} \geq \frac{C_0}{t^\beta}$ **then**
5:         $a(t) = \bar{a}(t)$
6:     **else**
7:         $a(t) = \tilde{a}(t)$
8:     **end if**
9:     Play arm $a(t)$, and observe reward $r_{a(t),t}$
10:     Update $X(t+1) = \lambda I_d + \sum_{j=1}^t x_{a(j),j} x_{a(j),j}^\top$, $\hat{\theta}_{t+1} = X(t+1)^{-1}(\sum_{j=1}^t x_{a(j),j} r_{a(j),j})$
11: **end for**

---

improve upon an incumbent. Here we define the incumbent as the largest posterior reward mean $r_t^+$ at time $t$. In Bayesian optimization, the incumbent is usually selected as the best reward value so far [11], or the largest reward mean so far [34]. In our setting with contextual bandits, we choose the latter for convenience in analysis. We also note that the EI form here can not be considered as an expected version of the TS in [3]. Even though the expected version of TS may be considered as $\mathbb{E}_{\mu \sim \mathcal{N}(\hat{\theta}_t, v_t^2 X(t)^{-1})} x_{i,t}^\top \mu$, but the expectation operator cannot be taken inside the *max* operator as it is a nonlinear function

**The closed form for EI.** Since $\mu \sim \mathcal{N}(\hat{\theta}_t, v^2 X(t)^{-1})$, the random variable $x_{i,t}^T \mu$ is Gaussian with mean $x_{i,t}^T \hat{\theta}_t$ and standard deviation $v s_{i,t}$, where we define $s_{i,t} = \sqrt{x_{i,t}^T X(t)^{-1} x_{i,t}}$. Setting $z_{i,t} = \frac{x_{i,t}^\top \hat{\theta}_t - r_t^+}{v s_{i,t}}$, we can express the expected improvement in closed form as follows

$$\alpha_{i,t}^{EI} = v s_{i,t}[z_{i,t} \Phi(z_{i,t}) + \phi(z_{i,t})], \tag{2}$$

where $\Phi(.)$ and $\phi(.)$ are the standard normal cdf and pdf function of the normal distribution respectively.

### 3.1 Proposed LinEI Algorithm.

We now use the above form of expected improvement to design our algorithm. The algorithm is performed iteratively as follows. At iteration $t$, it selects the arm suggested by the EI strategy if the expected improvement (measured by $\alpha_{i,t}^{EI}$ of all arms) is higher than a threshold. Otherwise it simply selects the arm suggested by the greedy strategy (corresponding to the arms whose posterior reward mean is $r_t^+$). The choice of thresholds plays an important role for convergence of our algorithm. We propose to use an adaptive threshold in the form of $\frac{C_0}{t^\beta}$ which is controlled by two parameters $C_0, \beta$. A relevant choice of parameters for convergence guarantee will be discussed in the next section. Our algorithm is summarized in Algorithm 1.

We remark that the proposed algorithm can be considered as a variant of EI strategy with a simple modification. This modification comes from our observation that at an iteration, when the EI value is less than some threshold, using a greedy strategy is better than the EI strategy in the sense that the instantaneous regret is smaller. This is our interesting observation about the relation between the EI and the greedy strategy, and we use this in our algorithm design. Without the modification, a pure EI algorithm can make the cumulative regret become unbounded in our theoretical analysis.

**Comparison with the LinUCB algorithm [1] and the LinTS algorithm [3].** LinUCB looks for the most optimistic value for an arm in an ellipsoid defined by a level set of the posterior rather than integrating over it, and then chooses an arm that maximizes the optimistic value. LinTS generates a sample $\mu$ from the posterior distributions $\mathcal{N}(\hat{\theta}_t, v_t^2 X(t)^{-1})$ of reward, and then chooses an arm that maximizes $x_{i,t}^T \mu$. In contrast, LinEI could choose an arm whose maximum optimistic value is lower unlike LinUCB, and/or choose an arm having lower $x_{i,t}^T \mu$ unlike LinTS but for which there are more possibilities for improvement.

## 3.2 Theoretical Analysis

In this section, we provide the regret bound for the proposed LinEI algorithm.

The EI technique is used so far in BAI and BO problems to seek the best arm with the highest reward mean. In such problems, the reward mean of each arm is fixed in advance. To leverage this property, [29] uses the *allocation matching* technique to allocate enough number of samples to suboptimal arms so that we can eliminate these arm with a high confidence. In BO, [11] uses the monotonicity property of incumbent to derive directly an upper bound for the simple regret. However, in contextual bandits where the best arm does not exist, and the mean reward of any arm can be different in different iterations, the monotonicity property of incumbent disappears, and the allocation technique is also no longer applicable. The proposed LinEI addresses these challenges. Moreover, it selects an arm based on not only EI strategy but also a greedy strategy depending on a threshold function. This makes our regret analysis different from the standard EI technique in [36],[27].

In our proof, we first extend several results in [11] in Bayesian optimization in a non-contextual setting to our setting with contexts and noises. For instance, the following lemma is used to upper bound and lower bound the expected improvement for each arm $i$.

**Lemma 3.1.** *Pick* $0 < \delta < 1$. *Set* $I_{i,t} = max\{0, x_{i,t}^T\theta^* - r_t^+\}$. *Then with probability* $1 - \frac{\delta}{t^2}$ *we have*

$$I_{i,t} - \beta_t s_{i,t} \leq \alpha_{i,t}^{EI} \leq I_{i,t} + (\beta_t + v_t)s_{i,t}$$

Here, $\beta_t = R\sqrt{d\ln(\frac{t^3}{\delta})} + 1$, $s_{i,t} = \sqrt{x_{i,t}^\top X(t)^{-1}x_{i,t}}$ and $v_t$ is used instead of $v$. Then, depending on the arm selection by our LinEI algorithm, we can upper bound the instantaneous regret $r_t = x_{a^*(t),t}^\top\theta^* - x_{a(t),t}^\top\theta^*$ by two different ways. Combining them, we achieve an upper bound for the instantaneous regret $r_t = x_{a^*(t),t}^\top\theta^* - x_{a(t),t}^\top\theta^*$ as in section C of our Supplementary Material:

$$r_t \leq \frac{\tau(\frac{\beta_t}{v_t})}{\tau(-\frac{\beta_t}{v_t})}(2\beta_t + v_t)s_{a(t),t} + \beta_t s_{a(t),t} + \frac{\tau(\frac{\beta_t}{v_t})}{\tau(-\frac{\beta_t}{v_t})}\frac{C_0}{t^\beta} + \sqrt{2\ln(Rt^\beta) + 2\sqrt{3C_0^{-2}d\ln(\frac{t}{\delta})}}v_t s_{a(t),t},$$

with probability at least $1 - \delta$. The function $\tau$ is defined as $\tau(z) = z\Phi(z) + \phi(z)$. Here, we note that parameter $v_t$ plays the role of parameter $v$ at time $t$ we discussed above. In our analysis, $v_t$ is used to eliminate the influence of $\beta_t$ so that $\frac{\beta_t}{v_t}$ is bounded as $t$ grows.

Finally, by a relevant choice of parameters $C_0$ and $\beta$, we achieve the regret bound for our proposed algorithm as follows:

**Theorem 3.2.** *Given any* $\delta \in (0,1)$. *If* $v_t = R\sqrt{3d\ln\frac{t}{\delta}} + 1$, $\sqrt{d} \leq C_0 \leq d$ *and* $0.5 \leq \beta \leq 3$ *then with probability* $1 - \delta$, *the cumulative regret of the LinEI algorithm is bounded as*

$$R(T) = \mathcal{O}(d\sqrt{T\ln^2(T)\ln\frac{T}{\delta}})$$

## 4 The NeuralEI algorithm for Neural Contextual Bandits

In this section, we extend our Algorithm 1 to the more general setting where the reward function is modelled by a fully connected neural network. Similar to the Neural Thompson Sampling approach [38], our algorithm maintains a Gaussian distribution for each arm's reward. At time $t$, the posterior distribution of the reward of arm $i$ is updated as follows. The mean is set to the output of the neural network, denoted by $f(x_{i,t}; \theta_{t-1})$, and the variance is defined as $\sigma_{i,t}^2 = \lambda g^\top(x_{i,t}; \theta_{t-1})U_{t-1}^{-1}g(x_{i,t}; \theta_{t-1})/m$, where the matrix $U_t^{-1}$ is updated as $U_t = U_{t-1} + g(x_{a(t),t}; \theta_t)g^\top(x_{a(t),t}; \theta_t)/m$ and parameter $\theta_t$ is the solution to the following minimization problem:

$$\min_\theta \sum_{i=1}^t [f(x_{a(i),i;\theta}) - r_{a(i),i}]^2/2 + m\lambda||\theta - \theta_0||_2^2/2, \tag{3}$$

where $\theta_0$ is randomly initialized network parameter. We can adapt gradient descent algorithms to solve this problem with step size $\eta$ and total number of iterations $J$ like the gradient descent algorithm of [39].

**Algorithm 2** Neural Expected Improvement Algorithm (NeuralEI)

---

**Input**: Number of rounds $T$, exploration variance $\nu$, network width $m$, regularization parameter $\lambda$ and parameters $C_0, \beta$

1: Set $U_0 = \lambda I$
2: **for** $t = 1$ to $T$ **do**
3:      Set $\overline{a}(t) := \text{argmax}_{i \in [K]} \alpha_{i,t}^{EI}$, $\tilde{a}(t) = \text{argmax}_{i \in [K]} \{f(x_{i,t}; \theta_{t-1})\}$
4:      **if** $\alpha_{\overline{a}(t),t}^{EI} \geq \frac{C_0}{t^\beta}$ **then**
5:         $a(t) = \overline{a}(t)$
6:      **else**
7:         $a(t) = \tilde{a}(t)$
8:      **end if**
9:      Play arm $a(t)$, and observe reward $r_{a(t),t}$
10:     Set $\theta_t$ to be the output of gradient descent for solving Eq(3)
11:     $U_t = U_{t-1} + g(x_{a(t),t}; \theta_t) g^\top(x_{a(t),t}; \theta_t)/m$
12: **end for**

---

**Expected Improvement for Neural Contextual Bandits.** We now define the form of the expected improvement in this setting. At each time step $t$, we denote $f_t^+ = \max_{i \in [K]} \{f(x_{i,t}; \theta_{t-1})\}$ which is the highest mean estimate of $f(x, \theta_{t-1})$ among all arms at time $t$. We define the expected improvement value of an arm $i$ at time $t$ as

$$\alpha_{i,t}^{EI} = \mathbb{E}_{\widetilde{f}_{i,k} \sim \mathcal{N}(f(x_{i,t}; \theta_{t-1}), \nu^2 \sigma_{i,t}^2)}[\max\{0, \widetilde{f}_{i,k} - f_t^+\}].$$

Further, by setting $z_{i,t} = \frac{f(x_{i,t}; \theta_{t-1}) - f_t^+}{\nu \sigma_{i,t}}$, the above expectation can be computed analytically as follows

$$\alpha_{i,t}^{EI} = \nu \sigma_{i,t}[z_{i,t} \Phi(z_{i,t}) + \phi(z_{i,t})] \tag{4}$$

Our NeuralEI algorithm is given in Algorithm 2. It starts by initializing $\theta_0 = (\text{vec}(W_1); ...; \text{vec}(W_L))$, where for each $1 \leq l \leq L - 1, W_l = (W, 0; 0, W)$, each entry of $W$ is generated independently from $N(0, 4/m)$; $W_L = (w^\top, -w^\top)$, each entry of $w$ is generated independently from $N(0, 2/m)$. NeuralEI extends our LinEI algorithm to the setting where the reward function $h$ is modelled by a fully connected neural network.

## 4.1 Regret Analysis

In this section, we provide a regret analysis of the NeuralEI algorithm. We first provide necessary background on the neural tangent kernel (NTK) theory, which plays an important role in our analysis. Following a recent line of research [39, 38], we define the covariance between two data point $x, y \in \mathbb{R}^d$ as follows: $\tilde{H}^{(1)}(x, y) = \sigma^{(1)}(x, y) = x^\top y$, $A^{(l)}(x, y) = \begin{pmatrix} \sigma^{(l)}(x, x) & \sigma^{(l)}(x, y) \\ \sigma^{(l)}(x, y) & \sigma^{(l)}(y, y) \end{pmatrix}$, $\sigma^{l+1}(x, y) = 2\mathbb{E}_{(u,v) \sim N(0, A^{(l)}(x,y))}[\sigma(u)\sigma(v)]$, $\tilde{H}^{(l+1)}(x, y) = 2\tilde{H}^{(l)}(x, y)\mathbb{E}_{(u,v) \sim N(0, A^{(l)}(x,y))}[\sigma'(u)\sigma'(v)] + \sigma^{(l+1)}(x, y)$. Similar to [39, 38], we assume that the number of rounds $T$ is *known* and denote the neural tangent kernel (NTK) matrix $H \in \mathbb{R}^{TK \times TK}$ based on all contextual vectors $\{x_{t,k}\}_{t \in [T], k \in [K]}$. Renumbering $\{x_{t,k}\}_{t \in [T], k \in [K]}$ as $\{x_i\}_{i=1,...,TK}$, then each entry $H_{ij}$ is defined as

$$H_{ij} = (\tilde{H}^{(L)}(x_i, x_j) + \sigma^{(L)}(x_i, x_j))/2, \tag{5}$$

for all $i, j \in [TK]$. Based on the above definition, we impose the following assumption on the contexts generated by the adversary and the corresponding NTK matrix $H$.

**Assumption 4.1.** Let $H$ be defined in Eq(5). There exists $\lambda_0 > 0$ such that $H \geq \lambda_0 I$. In addition, for any $t \in [T], k \in [K], \|x_{t,k}\|_2 = 1$ and $[x_{t,k}]_j = [x_{t,k}]_{j+d/2}$.

**Remark 1.** Compared to Algorithm 1 for linear bandits, our Algorithm 2 needs an additional Assumption 1 to guarantee the convergence. The assumption that the NTK matrix is positive definite has been considered in prior work on NTK which is a mild condition and also imposed in other

related works [4, 15, 39, 38]. The assumption on contexts ensures that $f(x_{i,t}; \theta_0) = 0$ for any $i \in [K], t \in [T]$.

The NTK technique builds a connection between deep neural networks and kernel methods. It enables us to adapt some complexity measures for kernel methods to describe the complexity of the neural network through the notation of the effective dimensions as defined in [39, 38]. The effective dimension $\tilde{d}$ of matrix $H$ with regularization parameter $\lambda$ is defined as $\tilde{d} = \frac{\log \det(I + H/\lambda)}{\log(1 + TK/\lambda)}$.

Using these notations, we are now ready to present the second main result of the paper. Let $a^*(t) = \text{argmax}_{i \in [K]} \mathbb{E}[r_{i,t}]$ be the optimal action at round $t$ that maximizes the expected reward, we define the *expected cumulative regret* after $T$ iterations as $\overline{R}(T) = \mathbb{E}[\sum_{t=1}^{T}(r_{a^*(t),t} - r_{a(t),t})]$. Then, we achieve the following upper regret bound for our Algorithm 2 by combining our EI techniques for LinEI with NTK techniques. A completed proof is provided in Supplementary Material.

**Theorem 4.2.** *Under Assumption 1, set the parameters in Algorithm 2 as* $\lambda = 1 + 1/T$, $\nu = B + R\sqrt{\tilde{d}\log(1 + TK/\lambda) + 2 + 2\log(1/\delta)}$, *where* $B = \max\{1, \sqrt{2h^\top H^{-1} h}\}$ *with* $h = (h(x_1), ..., h(x_{TK}))^\top$. *If* $\sqrt{\tilde{d}} \leq C_0 \leq \tilde{d}$, $\beta \geq 2$, *and the network width* $m$ *satisfies* $m \geq poly(\gamma, T, K, L, \log(1/\delta))$, *then with probability at least* $1 - \delta$, *the regret of Algorithm 2 is bounded as*

$$\overline{R}(T) \leq \mathcal{O}(\tilde{d}\sqrt{\beta\log(1 + TK)\log(T)T})$$

**Remark 2.** The regret bound depends on the parameter $\beta$. The best choice is $\beta = 2$ that tightens the regret. Theorem 4.2 implies the regret of NeuralEI is on the order of $\tilde{\mathcal{O}}(\tilde{d}\sqrt{T})$. This result matches the regret bound of NeuralUCB ([39]), NeuralTS ([38]), as well as of [12]. The effective dimension $\tilde{d}$ measures how quickly the eigenvalues of $H$ diminish, and only depends on $T$ logarithmically in several special cases ([35]), according to [39]. Furthermore, [38] shows that $\tilde{d}$ can be upper bounded if all contexts are nearly on some low-dimensional subspace of the RKHS space spanned by NTK. Currently, [20] gives an explicit sublinear regret bound for a neural network based UCB algorithm. However, their solution requires that contexts lie on on the d-dimensional hyper-sphere. Compared to these works, our EI based results may be of independent interest.

**Remark 3.** Similar to most of existing results in neural bandits, our results require a large value of $m$. This is rooted in the current deep learning theory based on the neural tangent kernel.

### 4.2 Technical Challenges

While the proof of NeuralEI follows the same lines as that of LinEI, we emphasize several key differences. First, we define filtration $\mathcal{F}_{t-1}$ as the union of history until time $t - 1$, and the contexts at time $t$. Given a time $t$, we define an event $\mathcal{E}_t^\sigma$ as follows:

$$\mathcal{E}_t^\sigma = \{\omega \in \mathcal{F}_t : \forall i \in [K], |\tilde{f}_{i,t} - f(x_{i,t}; \theta_{t-1})| \leq c_t \nu \sigma_{i,t}\},$$

where $c_t = \sqrt{4\log t + 2\log K}$.

Similarly, we define an event $\mathcal{E}_t^\mu$ as follows:

$$\mathcal{E}_t^\mu = \{\omega \in \mathcal{F}_t : \forall i \in [K], |f(x_{i,t}; \theta_{t-1}) - h(x_{t,k})| \leq \nu \sigma_{i,t} + \epsilon(m)\},$$

where $\epsilon(m)$ is defined as in [38].

Based on the definitions of event $\mathcal{E}_t^\sigma$ and $\mathcal{E}_t^\mu$, we achieve two different ways to lower bound and upper bound the expected improvement as follows:

**Lemma 4.3.** *Assume that the event* $\mathcal{E}_t^\mu$ *holds. Set* $J_{i,t} = \max\{0, h(x_{i,t}) - f_t^+\}$. *Then for every* $i \in [K]$, *we have*

$$J_{i,t} - \nu\sigma_{i,t} - \epsilon(m) \leq \alpha_{i,t}^{EI} \leq J_{i,t} + \nu\sigma_{i,t} + \epsilon(m).$$

**Lemma 4.4.** *Assume that the event* $\mathcal{E}_t^\sigma$ *holds. Set* $I_{i,t} = \max\{0, \tilde{f}_{i,t} - f_t^+\}$. *Then for every* $i \in [K]$, *we have*

$$I_{i,t} - c_t\nu\sigma_{i,t} \leq \alpha_{i,t}^{EI} \leq I_{i,t} + \nu(c_t + 1)\sigma_{i,t}.$$

While Lemma 4.3 is an adaptation of Lemma 3.1 to neural contextual bandits, Lemma 4.4 is a novel result compared to the regret analysis in our Section 3.1 for linear bandits. Challenges of analyzing the regret of NeuralEI come from the fact that there exists additionally a factor $\epsilon(m)$ in 4.3. Compared to LinEI, it is hard to analyze directly the regret $h(x_{a^*(t),t}) - h(x_{a(t),t})$. We divide the analysis into two cases: if $\sigma_{a^*(t),t} \leq \epsilon(m)$, we can follow the technique as in LinEI. Otherwise, we use another way to bound the regret using a novel result as we mentioned in the main paper. Please see Lemma D.11 in the appendix for details.

## 5 Related Works and Discussion

Given the vast literature on bandit algorithms, we restrict our review to linear bandits and neural contextual bandits.

**Linear Contextual Bandit.** A lower bound of $\Omega(d\sqrt{T})$ for linear bandits was given in [14], when the number of arms is allowed to be infinite. [1] analyze a UCB-style algorithm and provide a regret upper bound $\mathcal{O}(d\log(T)\sqrt{T} + \sqrt{dT\log(T/\delta)})$. Compared to this work, the regret of our proposed LinEI has an additional $\sqrt{\ln T}$, however, it still matches the information theoretic lower bound in this setting. When the number of arms $K$ is finite, [13] achieve a regret bound of $\mathcal{O}(\sqrt{Td\ln^3(KT\ln T)/\delta})$ with probability at least $1 - \delta$. [10] provides an algorithm based on exponential weights, with regret of order $\mathcal{O}(d\sqrt{T\log K})$. These algorithms may not be effective when the number of arms $K$ is large. For example, when $K$ is exponential in $d$, the regret bound of [13] would become $\tilde{\mathcal{O}}(d^2\sqrt{T})$ showing a quadratic growth in $d$.

The Thompson Sampling algorithm [3] and an alternative given in [2] bear an additional $\sqrt{d}$ in the regret bound. Very recently, [21] improved this regret bound of TS in some cases by integrating a doubly robust estimator with TS. However, this work requires additional significant computations and their setting is limited in independent contexts. In contrast, our EI-based algorithm achieves the optimal order of $d$ even in a general setting where the contexts may be controlled by an adaptive adversary.

Another approach for linear bandits is the Information Directed Sampling (IDS) which was introduced in [32]. It provides an action-selection mechanism by minimizing the information ratio between the squared expected regret and the mutual information between optimal action and the next observation over all action sampling distributions. IDS obtains a performance improvement over TS and UCB algorithms in some cases, but has heavy sampling requirements. It has been shown in their experiments that IDS requires significantly more compute time than Thompson sampling and UCB algorithms. Recently, [5] provided a modification of the arm scoring rule of IDS to reduce computations. However, both [32] and [5] only provide the bounds on *expected regret*. In contrast, our work provides regret bounds in terms of cumulative regret which is tighter than expected regret.

**Neural Contextual Bandit.** Neural contextual bandits are becoming attractive due to the current advancement in optimization and generalization of deep neural networks [4, 15]. Neural contextual bandits have been considered in both popular techniques UCB [39] and Thompson Sampling [38]. Currently, [6] proposes a novel neural exploration strategy and their solution achieves a sublinear regret on $T$. Compared to all existing works in this sub-field, our EI based algorithm is new, and may be of independent interest.

Due to space limit, we will add additional related works in Section B of Supplementary Material.

## 6 Experiments

### 6.1 Linear Bandits

In this subsection, we assess the performance of our LinEI algorithm on several benchmark datasets including `covertype`, `magic`, `avila`, `dry bean`, `statlog`, `letter`, `pendigits`, all from UCI [16]. See our Table for details. We compare the LinEI with methods designed for linear bandits including: LinTS [3], LinUCB [1], Linear Epsilon Greedy for the linear reward, LinIDS [32] for linear bandits. To transform these classification problems into multi-armed bandits, we adapt the

Table 1: Characteristics of benchmark datasets used in Section 6.

| Dataset | letter | pendigits | covertype | avila | magic | dry bean | statlog |
|---|---|---|---|---|---|---|---|
| Classes ($K$) | 26 | 10 | 7 | 12 | 2 | 7 | 7 |
| Feature Dimension | 17 | 16 | 54 | 10 | 10 | 16 | 8 |
| Dataset size | 20000 | 10992 | 581012 | 20867 | 19020 | 13611 | 58000 |

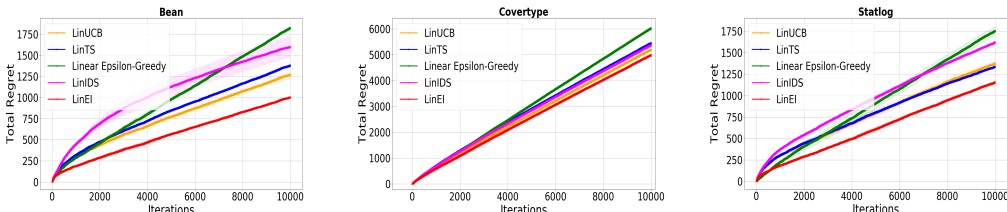

Figure 1: Comparison of our proposed LinEI and baseline algorithms in linear bandits.

disjoint models to build a context feature vector for each arm: given an input feature $x \in \mathbb{R}^d$ of a $k$-class classification problem, we build the context feature vector with dimension $kd$ as $x_1 = (x; 0; ...; 0), x_2 = (0; x; ...; 0), ..., x_k = (0; 0...; x)$. The algorithm generates a set of predicted reward and pulls the greedy arm. For these classification problems, if the algorithm selects a correct class by pulling the corresponding arm, it will receive a reward as 1, otherwise 0. The cumulative regret over time horizon $T$ is measured by the total mistakes made by the algorithm.

We set the time horizon of our algorithm to 10000 for all data sets. In the experiments, we shuffle all datasets randomly. For $(\lambda, \nu)$ used in LinUCB and LinTS and our algorithm, we set $\lambda = 1$ following previous works and do a grid search of $\nu \in \{1, 0.1, 0.01\}$ to select the parameter with the best performance. All experiments are repeated 10 times, and the average with standard error are reported. For LinIDS, we use the number of samples $M = 100$. For Linear Epsilon Greedy, we use $\epsilon = 0.1$. For our LinEI algorithm, we can choose any value $C_0 \in [\sqrt{d}, d]$ and $\beta \in [0.5, 3]$. For LinEI, we set $C_0 = \sqrt{d}$ and $\beta = 2$.

Figure 1 shows the total regret of all algorithms for datasets `bean`, `covertype` and `statlog`. The Linear Epsilon Greedy performs the worst. While LinUCB, LinTS and LinIDS are competitive, all these methods are significantly outperformed by the proposed algorithm. This difference in performance is because of the strategy used by each method. Due to space limit, the additional results on `magic`, `pendigits`, `letter`, and `avila` are shown in Section A of **Supplementary Material**. Our results on various datasets confirm that the exploration of the expected improvement is effective in practice. This also is widely observed in Bayesian optimization.

### 6.1.1 Comparison of LinUCB, LinTS and LinEI on a large-scale multi-label dataset

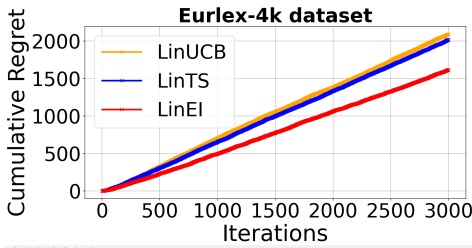

Figure 2: Comparison of LinUCB, LinTS and our proposed LinEI on a large-scale multi-label dataset.

We now compare our proposed LinEI against LinUCB and LinTS on the eurlex-4k dataset which is a large multi-label dataset [8]. It contains data with 5000 features and 3993 labels. It corresponds to a contextual bandit problem with 3993 actions and the dimension of context is $5000 + 3993 = 8993$ when mapped to our setting. This can be considered as a contextual bandit problem with large action

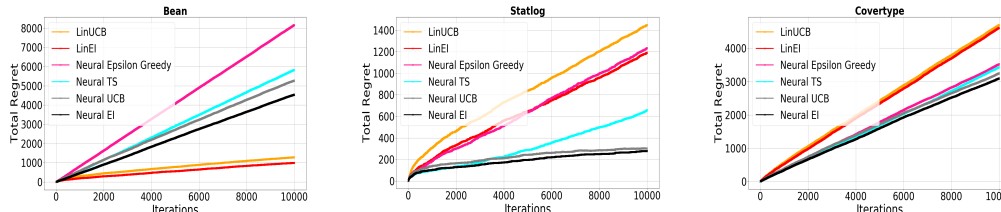

Figure 3: Comparison of NeuralEI and baseline algorithms on real-world datasets.

space and with very high dimensions. Due to the heavy computation, we run algorithms on only 3000 datapoints from the test set. Our experimental results show that LinEI scales better to high dimensions and to large action space (see our Figure 2). This can be explained as the regret of LinEI is better than that of LinTS by a factor $\sqrt{d}$ while LinUCB is an over- exploration strategy in high dimensions. In terms of required computations, LinEI is significantly cheaper than LinUCB and LinTS. Please see Section A.2 of the supplementary material for details. Due to space limit, we will add additional comparisons in term of dimension $d$ in Section A.1 of Supplementary Material.

### 6.2  Neural Bandits

We compare the proposed NeuralEI with baselines including: LinUCB [1], our LinEI for linear bandits problem, Neural Epsilon Greedy, NeuralUCB [39], NeuralTS [38]. We do the same classification problems as the experiments in subsection Linear Bandits. For methods using the neural network, we use one-hidden layer neural networks with 100 neurons to model the reward function. During posterior updating, gradient descent is run for 100 iterations with learning rate 0.001. For Neural UCB/Thompson Sampling and Neural EI, we use a grid search on $\lambda \in \{1, 10^1, 10^{-2}, 10^{-3}\}$ and $\nu \in \{10^{-1}, 10^{-2}, 10^{-3}, 10^{-4}, 10^{-5}\}$ as in [39] and [38]. We consider our algorithm on both synthetic datasets and real-world datasets. Due to space limit, we will provide results on synthetic datasets in Section A of Supplementary Material.

**Real-world Datasets.**   Similar to the subsection Linear Bandit, we build the context feature vector with dimension $kd$ as $x_1 = (x; 0; ...; 0), x_2 = (0; x; ...; 0), ..., x_k = (0; 0...; x)$. We also estimate our algorithm on datasets `bean`, `covertype` and `statlog`. Figure 3 show our results in the case of the neural bandits problem. Neural-based methods perform better because they can capture the nonlinearity of the underlying reward function. In real-datasets, while neural-based methods outperform LinEI and LinUCB for datasets `covertype` and `statlog`, these methods are not sample-efficient for learning the reward function of dataset `bean`. Perhaps, the reward function for dataset `bean` is linear. However, in all cases, our NeuralEI algorithm performs better than other neural-based methods. This suggests that using the expected improvement strategy is effective in both linear bandits and neural contextual bandits.

## 7  Conclusion

We introduced and formalized Expected Improvement as a new strategy for contextual bandits. We proposed two EI-based algorithms and analyzed them theoretically. The first algorithm assumes the reward function to be linear whilst the second algorithm is designed for the case when the reward function is general and can be modelled by a deep neural network. Our promising empirical results on real-world datasets suggested that our EI-based algorithms work well in practice compared to other approaches especially in high dimensions. We believe our work would be useful for further improvements and extensions. For example, extending EI to the reinforcement learning setting is an interesting open problem.

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
