# Supplementary Material

## A  Additional Experiments

As mentioned in the main paper, in this section, we provide several additional experiments for linear bandits on datasets: pendigit, avila, magic and letter. Figure 4 also showed that our proposed LinEI

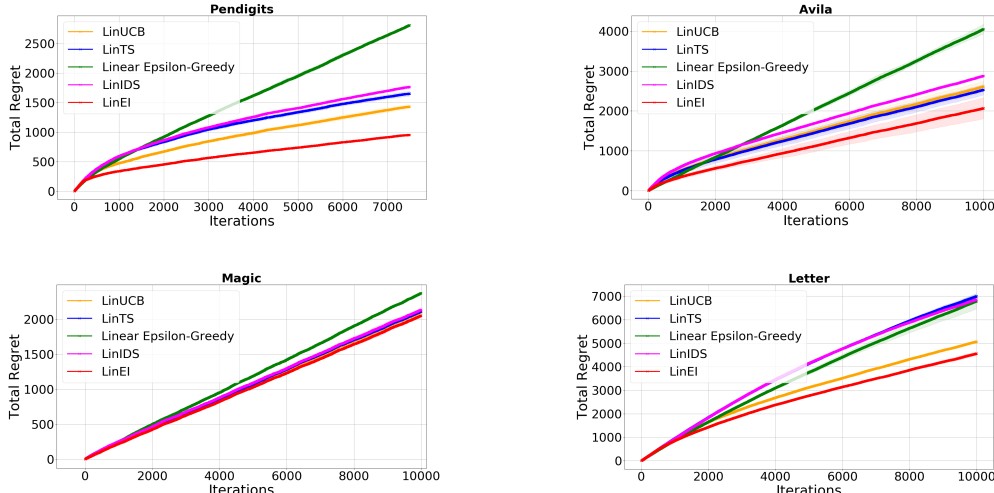

Figure 4: Comparison of LinEI and baseline algorithms on datasets.

algorithm outperforms LinUCB, LinTS and LinIDS and Linear Epsilon Greedy.

### A.1  Comparing LinEI with LinUCB and LinTS in terms of dimensions $d$

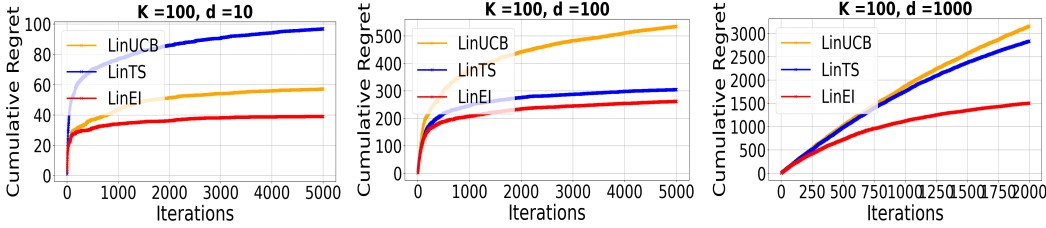

Figure 5: Comparison of our proposed LinEI and baseline algorithms with different dimensions.

Table 2: The average computation time (second)

| Dimension | LinUCB | LinTS | LinEI |
|-----------|--------|-------|-------|
| $d = 10$ | 0.018 | 0.015 | 0.024 |
| $d = 100$ | 0.46 | 0.73 | 0.12 |
| $d = 1000$ | 1.37 | 2.32 | 0.26 |

To demonstrate that our proposed LinEI scales to higt dimensions better than LinUCB, LinEI, we build a contextual bandit in linear setting as follows. We fix $K = 100$ actions. The context vectors are chosen uniformly at random from the unit ball. The reward function $r = x^T \theta^*$, where $\theta^*$ is generated *a priori*. Using this setting, we compared our LinEI with LinUCB and LinTS in a range of dimensions $d = 10, 100$ and $1000$. Our results show that

- LinEI scales to high dimensions better than LinUCB and LinTS. This can be explained as the regret of LinEI is better than that of LinTS by a factor $\sqrt{d}$ while LinUCB can be considered as an over-explored strategy in high dimensions. See our Figure 5.

- Concerning the computation time, LinUCB faces a NP-hard problem at each iteration [3], and thus spends more time for optimization especially in high dimensions. LinTS uses a random sampling based heuristic and requires a a multivariate Gaussian sampling step at each iteration. This is a high-dimensional multivariate Gaussian sampling problem [39]. It has computational cost and memory requirements which can rapidly become prohibitive in high dimension. In contrast, our LinEI can avoid both the NP-hard problem and the high-dimensional Gaussian sampling problem. Thus, the computation cost of LinEI is significantly cheaper in high dimensions. See our Table 2.

### A.2   Comparing LinEI with LinUCB and LinTS on an Large-scale Multi-Label Dataset

We now compare our proposed LinEI against LinUCB and LinTS on the eurlex-4k dataset which is a large multi-label dataset [8]. It contains data with 5000 features and 3993 labels. It corresponds to a contextual bandit problem with 3993 actions and the dimension of context is $5000 + 3993 = 8993$ when mapped to our setting. This can be considered as a contextual bandit problem with large action space and with very high dimensions. Due to the heavy computation, we run algorithms on only 3000 datapoints from the test set. Our experimental results show that LinEI scales better to high dimensions and to large action space (see our Figure 2). This can be explained as the regret of LinEI is better than that of LinTS by a factor $\sqrt{d}$ while LinUCB is an over- exploration strategy in high dimensions. In terms of required computations, LinEI is significantly cheaper than LinUCB and LinTS. The average-per iteration computation times of LinUCB, LinTS and LinEI are 1055.5s, 1496.2s and 181.3s respectively as shown in Table 3. This is because LinTS faces a high-dimensional multivariate Gaussian sampling problem. LinUCB requires to maximize quadratic forms at every round $1 \le t \le T$: argmax $x_{t,i}^T \theta$, where $i \in [K]$, and $\theta \in C_t$, where $C_t$ is the confident set at round $t$. We note that $\theta$ has the same dimension as the input $x_{t,i}$. This is an NP-hard problem as [1] has mentioned in the Related Work section. When the input dimension is high, it is expensive to find a maximum with a limited computational budget. In contrast, the proposed LinEI can avoid this NP-hard problem as long as finding argmax $\alpha_{i,t}^{EI}$, where $i \in [K]$ is solved. We can see that with a not very large value of $K$, this is no problem.

Table 3: The average computation time (second)

| Dataset | LinUCB | LinTS | LinEI |
|---|---|---|---|
| Eurlex-4k | 1055.5 | 1496.2 | 181.3 |

## B   Additional Related Works

**Bayesian optimization.**   EI has been widely used in Bayesian optimization which is a non-contextual problem. In contextual bandits we consider here, the mean reward of an arm is additionally associated with a context which may be controlled by an adversary. It follows that the mean reward for each arm can be different in each iteration depending on contexts, thus the notion of "best arm" does not exist. Due to this, the framework of EI as in Bayesian optimization does not naturally extend to contextual bandits. Moreover, our proposed algorithm in contextual bandits selects an arm based on not only EI strategy but also a greedy strategy depending on a threshold function. This makes our regret analysis different from the standard EI technique in [40],[29], and [37].

**Nonlinear Bandits**   The works of [38] and [23] extended linear bandits to kernel bandits and generalized linear bandits, respectively. However, they still required fairly restrictive assumptions on the reward function compared to neural contextual bandits. Neural contextual bandits are becoming attractive due to the current advancement in optimization and generalization of deep neural networks [4, 16]. Most of existing algorithms for neural bandits used either UCB technique [43, 21], or Thompson Sampling technique [42]. Currently, [6] proposed a novel neural exploration strategy and their solution achieved a sublinear regret on $T$. Compared to all existing works in this sub-field, our EI-based algorithm is new, and may be of independent interest.

**On the asymptotic regret.** In [24], authors analyzed the asymptotic regret for linear bandits. While showing that both optimism principle and Thompson sampling cannot be close to asymptotically optimal in the setting of linear bandits. Currently, the work in [36] analyzed the asymptotic regret for the contextual linear bandit problem which is similar to our setting. In their setting, context $x_t$ are drawn i.i.d. from a context distribution. Different from these settings, we consider contextual linear bandits where contexts may be controlled by an adversary. Further, we also consider the neural contextual bandits which are more general than the linear bandit considered in [24] and [36]. To our knowledge, the analysis of the asymptotic regret for neural contextual bandits is still an open problem.

## C Regret Analysis for LinEI Algorithm

### C.1 Auxiliary Results

For our theoretical analysis for LinEI, we use following several auxiliary lemmas.

**Lemma C.1.** *(Upper Bounds of a Hyperharmonic Series, [12]) Given a hyperharmonic series* $p_n = \sum_{t=1}^{n} \frac{1}{t^\beta}$, *where* $n \in \mathbb{N}$. *Then,*

- $p_n < 1 + \frac{n^{1-\beta}-1}{1-\beta}$ *if* $\beta \geq 0$ *and* $\beta \neq 1$,

- $p_n < 1 + ln(n)$ *if* $\beta = 1$

**Lemma C.2** (Lemma 1 of [3]). *For all t,* $0 < \delta < 1$, *with probability* $1 - \frac{\delta}{t^2}$, *we have*

$$\forall i : |x_{i,t}^T \hat{\theta}_t - x_{i,t}^T \theta^*| \leq \beta_t s_{i,t},$$

*where* $\beta_t = R\sqrt{dln(\frac{t^3}{\delta})} + 1$, *and* $s_{i,t} = \sqrt{x_{i,t}^T X(t)^{-1} x_{i,t}}$.

**Lemma C.3** (Lemma 3 of [14]). *For* $T \geq 2$, *we have*

$$\sum_{i=1}^{T} s_{a(t),t} \leq 5\sqrt{dTln(T)}.$$

We note that except that Lemma C.2 and Lemma C.3 are used from the analysis of LinTS and LinUCB, the following results are different from those of LinUCB and LinTS.

To derive the regret for our LinEI algorithm, we will need a lower bound and a upper bound of EI for each arm $i \in [K]$ as follows.

**Lemma C.4.** *Pick* $0 < \delta < 1$. *Set* $I_{i,t} = max\{0, x_{i,t}^T \theta^* - r_t^+\}$. *Then with probability* $1 - \frac{\delta}{t^2}$ *we have*

$$I_{i,t} - \beta_t s_{i,t} \leq \alpha_{i,t}^{EI} \leq I_{i,t} + (\beta_t + v_t)s_{i,t}.$$

*Proof.* If $s_i(t) = 0$ then $\alpha_{i,t}^{EI} = I_{i,t}$, which makes the result trivial. We now assume that $s_{i,t} > 0$. Set $q = \frac{x_{i,t}^T \theta^* - r_t^+}{s_{i,t}}$ and $u = \frac{x_{i,t}^T \hat{\theta}_t - r_t^+}{s_{i,t}}$. Set $\tau(z) = z\Phi(z) + \phi(z)$. Then we have that

$$\alpha_{i,t}^{EI} = v_t s_{i,t} \tau(\frac{u}{v_t}).$$

By Lemma C.2, we have that $|u - q| \leq \beta_t$ with probability $1 - \delta$. As $\tau'(z) = \Phi(z) \in [0, 1]$, $\tau$ is non-decreasing and $\tau(z) \leq 1 + z$ for $z > 0$. Hence,

$$
\begin{aligned}
\alpha_{i,t}^{EI} &\leq v_t s_{i,t} \tau(\frac{max\{0, q\} + \beta_t}{v_t}) \\
&\leq v_t s_{i,t}(\frac{max\{0, q\} + \beta_t}{v_t} + 1) \\
&= I_{i,t} + (\beta_t + v_t)s_{i,t}
\end{aligned}
$$

If $I_{i,t} = 0$ then the lower bound is trivial as $\alpha_{i,t}^{EI}$ is non-negative. Thus suppose $I_{i,t} > 0$. Since $\alpha_{i,t}^{EI} \geq 0$ and $\tau(z) \geq 0$ for all $z$, and $\tau(z) = z + \tau(-z) \geq z$. Therefore,

$$
\begin{aligned}
\alpha_{i,t}^{EI} &\geq v_t s_{i,t} \tau(\frac{q - \beta_t}{v_t}) \\
&\geq v_t s_{i,t}(\frac{q - \beta_t}{v_t}) \\
&= I_{i,t} - \beta_t s_{i,t}
\end{aligned}
$$

$\square$

## C.2   Proof for Theorem 3.2

We now derive an upper bound for the cumulative regret $R(T)$. To do this, we upper bound $r_t = x_{a^*(t),t}^T \theta^* - x_{a(t),t}^T \theta^*$ for every $t \in [T]$. We break down $r_t$ into two terms as follows:

$$
\begin{aligned}
r_t &= x_{a^*(t),t}^T \theta^* - x_{a(t),t}^T \theta^* \\
&= \underbrace{x_{a^*(t),t}^T \theta^* - r_t^+}_{\text{Term 1}} + \underbrace{r_t^+ - x_{a(t),t}^T \theta^*}_{\text{Term 2}},
\end{aligned}
$$

where $r_t^+ = \max_{i \in [K]}\{x_{i,t}^T \hat{\theta}_t\}$. We will upper bound Term 1 and Term 2 according to two different cases of $a(t)$, either $a_t = \operatorname{argmax}_{i \in \mathcal{K}} \alpha_{i,t}^{EI}$, or $a_t = \operatorname{argmax}_{i \in \mathcal{K}}\{x_{i,t}^\top \hat{\theta}_t\}$

## C.3   Considering the case $a_t = \operatorname{argmax}_{i \in \mathcal{K}} \alpha_{i,t}^{EI}$

We note that this case happens when $\alpha_{\bar{a}(t),t}^{EI} \geq \frac{C_0}{t^\beta}$.

**Bounding Term 1**

**Lemma C.5.** *Pick $\delta \in (0,1)$. Then with probability at least $1 - \frac{\delta}{t^2}$ we have*

$$
x_{a^*(t),t}^T \theta^* - r_t^+ \leq \frac{\tau(\frac{\beta_t}{v_t})}{\tau(-\frac{\beta_t}{v_t})}(2\beta_t + v_t)s_{a(t),t}.
$$

*Proof.* First, we consider $s_{a^*(t),t} > 0$. If $x_{a^*(t),t}^T \theta^* < r_t^+$ then the lemma will be trivial. We now consider $x_{a^*(t),t}^T \theta^* \geq r_t^+$. Following the derivation of the acquisition function $\alpha^{EI}$, we have $\alpha_{a^*(t),t}^{EI} = v_t s_{a^*(t),t} \tau(\frac{x_{a^*(t),t}^T \hat{\theta}_t - r_t^+}{v_t s_{a^*(t),t}})$. Further, we also have $\frac{x_{a^*(t),t}^T \hat{\theta}_t - r_t^+}{v_t s_{a^*(t),t}} \geq \frac{-\beta_t}{v_t}$ with probability $1 - \frac{\delta}{t^2}$. It is because $x_{a^*(t),t}^T \hat{\theta}_t - x_{a^*(t),t}^T \theta^* \geq -\beta_t s_{a^*(t),t}$ with probability $1 - \frac{\delta}{t^2}$ and we are considering the case when $x_{a^*(t),t}^T \theta^* \geq r_t^+$. Therefore, $\alpha_{a^*(t),t}^{EI} \geq v_t \tau(\frac{-\beta_t}{v_t})s_{a^*(t),t}$ with probability $1 - \frac{\delta}{t^2}$.

Now, we combine inequalities $\alpha_{a^*(t),t}^{EI} \geq v_t \tau(\frac{-\beta_t}{v_t})s_{a^*(t),t}$ and $\alpha_{a^*(t),t}^{EI} \geq I_{a^*(t),t} - \beta_t s_{a^*(t),t}$ which is proven in Lemma C.4, we obtain the following inequality:

$$
I_{a^*(t),t} \leq \frac{\tau(\frac{\beta_t}{v_t})}{\tau(-\frac{\beta_t}{v_t})}\alpha_{a^*(t),t}^{EI} \tag{6}
$$

Here we use the fact $\tau(z) = z + \tau(-z)$ for $z = \frac{\beta_t}{v_t}$. Thus, with probability at least $1 - \frac{\delta}{t^2}$ we achieve

$$
\begin{aligned}
x_{a^*(t),t}^T \theta^* - r_t^+ &\leq I_{a^*(t),t} \\
&\leq \frac{\tau(\frac{\beta_t}{v_t})}{\tau(-\frac{\beta_t}{v_t})}\alpha_{a^*(t),t}^{EI}
\end{aligned}
$$

$$\leq \frac{\tau(\frac{\beta_t}{v_t})}{\tau(-\frac{\beta_t}{v_t})}\alpha_{a(t),t}^{EI}$$

$$\leq \frac{\tau(\frac{\beta_t}{v_t})}{\tau(-\frac{\beta_t}{v_t})}(\max\{0, x_{a(t),t}^T\hat{\theta}_t - r_t^+\} + (\beta_t + v_t)s_{a(t),t})$$

$$\leq \frac{\tau(\frac{\beta_t}{v_t})}{\tau(-\frac{\beta_t}{v_t})}(\max\{0, x_{a(t),t}^T\hat{\theta}_t + \beta_t s_{a(t),t} - r_t^+\} + (\beta_t + v_t)s_{a(t),t})$$

$$\leq \frac{\tau(\frac{\beta_t}{v_t})}{\tau(-\frac{\beta_t}{v_t})}(\max\{0, \beta_t s_{a(t),t}\} + (\beta_t + v_t)s_{a(t),t})$$

$$= \frac{\tau(\frac{\beta_t}{v_t})}{\tau(-\frac{\beta_t}{v_t})}(2\beta_t + v_t)s_{a(t),t},$$

where the first inequality holds by the definition of the function $I_t$. The second one comes from Eq(9). The third one holds by the property of the chosen point $a(t) = \text{argmax}_{i\in[K]}\alpha_{i,t}^{EI}$. The fourth inequality holds due to Lemma C.2. The sixth inequality holds due to the fact that $x_{a(t),t}^T\hat{\theta}_t \leq r_t^+$.

If $s_{a^*(t),t} = 0$ then by definition of $\alpha_{a^*(t),t}^{EI}$, we have $\alpha_{a^*(t),t}^{EI} = I_{a^*(t),t}$. We have $I_{a^*(t),t} = \alpha_{a^*(t),t}^{EI} \leq \alpha_{a(t),t}^{EI}$, where we use the definition $\alpha_{a(t),t}^{EI} = \max_{i\in[K]}\alpha_{i,t}^{EI}$. Similar to the above proof, we obtain $x_{a^*(t),t}^T\theta^* - r_t^+ \leq (2\beta_t + v_t)s_{a(t),t} \leq \frac{\tau(\frac{\beta_t}{v_t})}{\tau(-\frac{\beta_t}{v_t})}(2\beta_t + v_t)s_{a(t),t}$ because $\frac{\tau(\frac{\beta_t}{v_t})}{\tau(-\frac{\beta_t}{v_t})} \geq \frac{\tau(0)}{\tau(0)} = 1$.
Thus, the lemma holds. $\square$

### Bounding Term 2

**Lemma C.6.** *Pick a $\delta \in (0,1)$. If $v_t = R\sqrt{3d\ln(\frac{t}{\delta})} + 1$, then with probability $1 - \frac{\delta}{t^2}$ we have*

$$r_t^+ - x_{a(t),t}^T\theta^* \leq \sqrt{2\ln(Rt^\beta) + 2\ln(\sqrt{3C_0^{-2}d\ln(\frac{t}{\delta})} + C_0^{-1}R^{-1})}v_t s_{a(t),t} + \beta_t s_{a(t),t}.$$

*Proof.* We have

$$\alpha_{a(t),t}^{EI} = (x_{a(t),t}^T\hat{\theta}_t - r_t^+)\Phi(\frac{x_{a(t),t}^T\hat{\theta}_t - r_t^+}{v_t s_{a(t),t}}) + v_t s_{a(t),t}\phi(\frac{x_{a(t),t}^T\hat{\theta}_t - r_t^+}{v_t s_{a(t),t}})$$

$$\leq v_t s_{a(t)}(t)\phi(\frac{x_{a(t),t}^T\hat{\theta}_t - r_t^+}{v_t s_{a(t),t}})$$

$$= v_t s_{a(t),t}\frac{1}{2\sqrt{\pi}}\exp(-\frac{1}{2}(\frac{x_{a(t),t}^T\hat{\theta}_t - r_t^+}{v_t s_{a(t),t}})^2),$$

where the first inequality holds due to the definition of action $a(t)$. The second equality holds due to the definition of $\alpha_{a(t),t}^{EI}$. The second inequality comes from the fact that $x_{a(t),t}^T\hat{\theta}_t \leq r_t^+$. The third equality holds due to the definition of function $\phi(.)$.

From the last inequality, we obtain

$$|x_{a(t),t}^T\hat{\theta}_t - r_t^+| \leq \sqrt{2\ln(\frac{v_t s_{a(t),t}}{\alpha_{a(t),t}^{EI}})}v_t s_{a(t),t}.$$

By applying the facts $\alpha_{a(t),t}^{EI} \geq \frac{C_0}{t^\beta}$, $s_{a(t),t} \leq 1$, and $v_t = R\sqrt{3d\ln(\frac{t}{\delta})} + 1$ to the above inequality, we have

$$|x_{a(t),t}^T\hat{\theta}_t - r_t^+| \leq \sqrt{2\ln(\frac{(R\sqrt{3d\ln(\frac{t}{\delta})} + 1)t^\beta}{C_0})}v_t s_{a(t),t} = \sqrt{2\ln(Rt^\beta) + 2\ln(\sqrt{3C_0^{-2}d\ln(\frac{t}{\delta})} + C_0^{-1}R^{-1})}v_t s_{a(t),t}.$$

Finally, we have

$$
\begin{aligned}
r_t^+ - x_{a(t),t}^T \theta^* &= (r_t^+ - x_{a(t),t}^T \hat{\theta}_t) + (x_{a(t),t}^T \hat{\theta}_t - x_{a(t),t}^T \theta^*) \\
&\leq \sqrt{2\ln(Rt^\beta) + 2\ln(\sqrt{3C_0^{-2}d\ln(\frac{t}{\delta})} + C_0^{-1}R^{-1})} v_t s_{a(t),t} + \beta_t s_{a(t),t},
\end{aligned}
$$

where the inequality holds by using Lemma C.2. $\qquad\square$

## C.4 Considering the case $a(t) = \mathbf{argmax}_{i\in[K]}\{x_{i,t}^T \hat{\theta}_t\}$

**Bounding Term 1**

**Lemma C.7.** *Pick $\delta \in (0,1)$. Then with probability at least $1 - \frac{\delta}{t^2}$ we have*

$$
x_{a^*(t),t}^T \theta^* - r_t^+ \leq \frac{\tau(\frac{\beta_t}{v_t})}{\tau(-\frac{\beta_t}{v_t})} \frac{C_0}{t^\beta}.
$$

*Proof.* If $s_{a^*(t),t} = 0$ then by definition of $\alpha_{a^*(t),t}^{EI}$, $\alpha_{a^*(t),t}^{EI} = I_{a^*(t),t}$. We have $I_{a^*(t),t} = \alpha_{a^*(t),t}^{EI} \leq \alpha_{a(t),t}^{EI} \leq \frac{C_0}{t^\beta}$.

We now consider $s_{a^*(t),t} > 0$. If $x_{a^*(t),t}^T \theta^* < r_t^+$ then the lemma will be trivial. We now consider $x_{a^*(t),t}^T \theta^* \geq r_t^+$. Following the derivation of the acquisition function $\alpha^{EI}$, we have $\alpha_{a^*(t),t}^{EI} = v_t s_{a^*(t),t} \tau(\frac{x_{a^*(t),t}^T \hat{\theta}_t - r_t^+}{v_t s_{a^*(t),t}})$. Further, we also have $\frac{x_{a^*(t),t}^T \hat{\theta}_t - r_t^+}{v_t s_{a^*(t),t}} \geq \frac{-\beta_t}{v_t}$ with probability $1 - \frac{\delta}{t^2}$. It is because $x_{a^*(t),t}^T \hat{\theta}_t - x_{a^*(t),t}^T \theta^* \geq -\beta_t s_{a^*(t),t}$ with probability $1 - \frac{\delta}{t^2}$ and we are considering the case when $x_{a^*(t),t}^T \theta^* \geq r_t^+$. Therefore, $\alpha_{a^*(t),t}^{EI} \geq v_t \tau(\frac{-\beta_t}{v_t}) s_{a^*(t),t}$ with probability $1 - \frac{\delta}{t^2}$.

Now, we combine inequalities $\alpha_{a^*(t),t}^{EI} \geq v_t \tau(\frac{-\beta_t}{v_t}) s_{a^*(t),t}$ and $\alpha_{a^*(t),t}^{EI} \geq I_{a^*(t),t} - \beta_t s_{a^*(t),t}$ which is proven in Lemma C.4, we obtain the following inequality:

$$
I_{a^*(t),t} \leq \frac{\tau(\frac{\beta_t}{v_t})}{\tau(-\frac{\beta_t}{v_t})} \alpha_{a^*(t),t}^{EI} \tag{7}
$$

Here we use the fact $\tau(z) = z + \tau(-z)$. Finally, with probability at least $1 - \frac{\delta}{t^2}$ we achieve

$$
\begin{aligned}
x_{a^*(t),t}^T \theta^* - r_t^+ &\leq I_{a^*(t),t} \\
&\leq \frac{\tau(\frac{\beta_t}{v_t})}{\tau(-\frac{\beta_t}{v_t})} \alpha_{a^*(t),t}^{EI} \\
&\leq \frac{\tau(\frac{\beta_t}{v_t})}{\tau(-\frac{\beta_t}{v_t})} \alpha_{\bar{a}(t),t}^{EI} \\
&\leq \frac{\tau(\frac{\beta_t}{v_t})}{\tau(-\frac{\beta_t}{v_t})} \frac{C_0}{t^\beta}
\end{aligned}
$$

where the first inequality holds by the definition of the function $I_t$. The second one comes from Eq(7). The third one holds by the definition of $\bar{a}(t)$. The fourth one holds because $\alpha_{\bar{a}(t),t}^{EI} \leq \frac{C_0}{t^\beta}$. $\qquad\square$

**Bounding Term 2**

**Lemma C.8.** *Pick a $\delta \in (0,1)$. Then with probability $1 - \frac{\delta}{t^2}$ we have*

$$
r_t^+ - x_{a(t),t}^T \theta^* \leq \beta_t s_{a(t),t}.
$$

*Proof.* By definition $a(t) = \text{argmax}_{i \in [K]} \{x_{i,t}^T \hat{\theta}_t\}$, we have $r_t^+ = x_{a(t),t}^T \hat{\theta}_t$. By Lemma C.2, we have $r_t^+ - x_{a(t),t}^T \theta^* \leq \beta_t s_{a(t),t}$. The lemma holds. $\qquad\square$

We now are ready to bound $x_{a^*(t),t}^T \theta^* - x_{a(t),t}^T \theta^*$ for both cases of $a(t)$.

**Lemma C.9.** *For every $1 \leq t \leq T$, with probability $1 - \frac{\delta}{t^2}$ we always have*

$$x_{a^*(t),t}^T \theta^* - x_{a(t),t}^T \theta^* \leq \frac{\tau(\frac{\beta_t}{v_t})}{\tau(-\frac{\beta_t}{v_t})}(2\beta_t + v_t)s_{a(t),t} + \sqrt{2ln(Rt^\beta) + 6\sqrt{C_0^{-2}dln(\frac{t+1}{\delta})}} v_t s_{a(t),t} +$$

$$+ \beta_t s_{a(t),t} + \frac{\tau(\frac{\beta_t}{v_t})}{\tau(-\frac{\beta_t}{v_t})} \frac{C_0}{t^\beta}$$

*Proof.* Following Algorithm 1, we consider two cases of $a(t)$.

- if $a(t) = \text{argmax}_{i \in \mathcal{K}} \alpha_{i,t}^{EI}$, then by combining Lemma C.5 and C.6 we have

$$x_{a^*(t),t}^T \theta^* - x_{a(t),t}^T \theta^* = (x_{a^*(t),t}^T \theta^* - r_t^+) + (r_t^+ - x_{a(t),t}^T \theta^*)$$

$$\leq \frac{\tau(\frac{\beta_t}{v_t})}{\tau(-\frac{\beta_t}{v_t})}(2\beta_t + v_t)s_{a(t),t}$$

$$+ \sqrt{2ln(Rt^\beta) + 2ln(\sqrt{3C_0^{-2}dln(\frac{t}{\delta})} + C_0^{-1}R^{-1})} v_t s_{a(t),t} + \beta_t s_{a(t),t}$$

- if $a(t) = \text{argmax}_{i \in [K]} \{x_{i,t}^T \hat{\theta}_t\}$, the by combining Lemma C.7 and C.8 we have

$$x_{a^*(t),t}^T \theta^* - x_{a(t),t}^T \theta^* = (x_{a^*(t),t}^T \theta^* - r_t^+) + (r_t^+ - x_{a(t),t}^T \theta)$$

$$\leq \frac{\tau(\frac{\beta_t}{v_t})}{\tau(-\frac{\beta_t}{v_t})} \frac{C_0}{t^\beta} + \beta_t s_{a(t),t}$$

For both cases of $a_t$, we always obtain the following upper bound for $x_{a^*(t),t}^T \theta^* - x_{a(t),t}^T \theta^*$ as

$$x_{a^*(t),t}^T \theta^* - x_{a(t),t}^T \theta^* \leq \frac{\tau(\frac{\beta_t}{v_t})}{\tau(-\frac{\beta_t}{v_t})}(2\beta_t + v_t)s_{a(t),t} + \sqrt{2ln(Rt^\beta) + 2ln(\sqrt{3C_0^{-2}dln(\frac{t}{\delta})} + C_0^{-1}R^{-1})} v_t s_{a(t),t} +$$

$$+ \beta_t s_{a(t),t} + \frac{\tau(\frac{\beta_t}{v_t})}{\tau(-\frac{\beta_t}{v_t})} \frac{C_0}{t^\beta}$$

Using $v_t = \beta_t = R\sqrt{dln(\frac{t^3}{\delta})} + 1$, we obtain

$$x_{a^*(t),t}^T \theta^* - x_{a(t),t}^T \theta^* \leq 3\frac{\tau(1)}{\tau(-1)}\beta_t s_{a(t),t} + \sqrt{2ln(Rt^\beta) + 2ln(\sqrt{3C_0^{-2}dln(\frac{t}{\delta})} + C_0^{-1}R^{-1})}\beta_t s_{a(t),t} +$$

$$+ \beta_t s_{a(t),t} + \frac{\tau(1)}{\tau(-1)} \frac{C_0}{t^\beta}$$

$$\qquad\square$$

We now are ready to prove the correctness of Theorem 3.2.

**Theorem C.10.** *Given any $\delta \in (0, 1)$. If $\sqrt{d} \leq C_0 \leq d$ and $0.5 \leq \beta \leq 3$ then with probability $1 - \delta$, the cumulative regret of the Expected Improvement algorithm is bounded as*

$$R(T) = \mathcal{O}(d\sqrt{Tln^2(T)ln\frac{T}{\delta}}).$$

*Proof.* Using Lemma, for every $1 \le t \le T$, we have

$$x_{a^*(t),t}^T \theta^* - x_{a(t),t}^T \theta^* \le 3\frac{\tau(1)}{\tau(-1)}\beta_t s_{a(t),t} + \sqrt{2\ln(Rt^\beta) + 2\ln(\sqrt{3C_0^{-2}d\ln(\frac{t}{\delta})} + C_0^{-1}R^{-1})}\beta_t s_{a(t),t} +$$

$$+ \beta_t s_{a(t),t} + \frac{\tau(1)}{\tau(-1)}\frac{C_0}{t^\beta}$$

Thus,

$$
\begin{aligned}
R(T) &= \sum_{t=1}^{T}(x_{a^*(t),t}^T \theta^* - x_{a(t),t}^T \theta^*) \\
&\le \sum_{t=1}^{T}[3\frac{\tau(1)}{\tau(-1)}\beta_t s_{a(t),t} + \sqrt{2\ln(Rt^\beta) + 2\ln(\sqrt{3C_0^{-2}d\ln(\frac{t}{\delta})} + C_0^{-1}R^{-1})}\beta_t s_{a(t),t} + \\
&\quad + \beta_t s_{a(t),t} + \frac{\tau(1)}{\tau(-1)}\frac{C_0}{t^\beta}] \\
&= \sum_{t=1}^{T}[3\frac{\tau(1)}{\tau(-1)} + \sqrt{2\ln(Rt^\beta) + 2\ln(\sqrt{3C_0^{-2}d\ln(\frac{t}{\delta})} + C_0^{-1}R^{-1})} + 1]\beta_t s_{a(t),t} + \sum_{t=1}^{T}\frac{\tau(1)}{\tau(-1)}\frac{C_0}{t^\beta} \\
&\le [3\frac{\tau(1)}{\tau(-1)} + \sqrt{2\ln(RT^\beta) + 2\ln(\sqrt{3C_0^{-2}d\ln(\frac{T}{\delta})} + C_0^{-1}R^{-1})} + 1]\beta_T \sum_{t=1}^{T} s_{a(t),t} + \sum_{t=1}^{T}\frac{\tau(1)}{\tau(-1)}\frac{C_0}{t^\beta} \\
&\le 5[3\frac{\tau(1)}{\tau(-1)} + \sqrt{2\ln(RT^\beta) + 2\ln(\sqrt{3C_0^{-2}d\ln(\frac{T}{\delta})} + C_0^{-1}R^{-1})} + 1]\beta_T \sqrt{dT\ln(T)} + \frac{\tau(1)}{\tau(-1)}\sum_{t=1}^{T}\frac{C_0}{t^\beta},
\end{aligned}
$$

where in the second inequality, we use the facts that $\beta_t \le \beta_T$, $v_t \le v_T$ as $t \le T$. In the third inequality, we use Lemma C.3. We consider two cases:

- if $\beta = 1$ then by Lemma C.1, $\sum_{t=1}^{T}\frac{1}{t^\beta} \le 1 + \ln T$. Hence, using the assumption that $\sqrt{d} \le C_0 \le d$, we get that $C_0^{-2}d \le 1$ and $C_0 \sum_{t=1}^{T}\frac{1}{t^\beta} \le d(1 + \ln T)$. We recall that $\beta_T = R\sqrt{d\ln(\frac{T^3}{\delta})} + 1$, and $\tau(1), \tau(-1)$ are constant. Thus, $R(T) = \mathcal{O}(d\sqrt{T\ln^2(T)\ln\frac{T}{\delta}})$.

- if $\beta \ne 1$ then by Lemma C.1, $\sum_{t=1}^{T}\frac{1}{t^\beta} \le 1 + \frac{T^{1-\beta}-1}{1-\beta}$. Combining with the assumption that $1/2 \le \beta \le 3$, we consider two cases:

  - if $1/2 \le \beta < 1$, we have $\sum_{t=1}^{T}\frac{1}{t^\beta} \le 1 + \frac{T^{1-\beta}-1}{1-\beta} \le 1 + 2(\sqrt{T} - 1)$. Hence, $C_0 \sum_{t=1}^{T}\frac{1}{t^\beta} \le d(1 + 2(\sqrt{T} - 1))$. Thus, $R(T) = \mathcal{O}(d\sqrt{T\ln^2(T)\ln\frac{T}{\delta}})$.

  - if $1 < \beta \le 3$, then $\sum_{t=1}^{T}\frac{1}{t^\beta} \le 1 + \frac{T^{1-\beta}-1}{1-\beta} \le 1$ and $T^\beta \le T^3$. Thus, $R(T) = \mathcal{O}(d\sqrt{T\ln^2(T)\ln\frac{T}{\delta}})$.

$\square$

**Remark.** We choose $C_0 \in [\sqrt{d}, d]$ to eliminate the term $d$ in the expression $\sqrt{C_0^{-2}d\ln(\frac{T}{\delta})}$ and to ensure that the term $\sum_{i=1}^{T}\frac{C_0}{t^\beta} = \mathcal{O}(d\sqrt{T})$. We choose $\beta \le 3$ to ensure that the order of $R(T)$ is the same as the case when $1/2 \le \beta < 1$. For $\beta > 3$, the higher $\beta$, the larger the cumulative regret is.

## D   Regret Analysis for NeuralEI Algorithm

In this section, we provide the regret analysis for the NeuralEI algorithm. We start with the definition of the filtration $\mathcal{F}_{t-1}$ as the union of history until time $t - 1$, and the contexts at time $t$. For any $t$, we

define an event $\mathcal{E}_t^\sigma$ as follows:

$$\mathcal{E}_t^\sigma = \{\omega \in \mathcal{F}_t : \forall i \in [K], \ |\tilde{f}_{i,t} - f(x_{i,t}; \theta_{t-1})| \le c_t \nu \sigma_{i,t}\},$$

where $c_t = \sqrt{4\mathrm{log}t + 2\mathrm{log}K}$.

For any $t$, we define an event $\mathcal{E}_t^\mu$ as follows:

$$\mathcal{E}_t^\mu = \{\omega \in \mathcal{F}_t : \forall i \in [K], \ |f(x_{i,t}; \theta_{t-1}) - h(x_{t,k})| \le \nu \sigma_{i,t} + \epsilon(m)\},$$

where $\epsilon(m)$ is defined as in [42]:

$$\epsilon(m) = \bar{C}_1 T^{2/3} m^{-1/6} \lambda^{-2/3} L^3 \sqrt{\log m} + \bar{C}_2 (1 - \eta \, m\lambda)^J \sqrt{TL/\lambda}$$
$$+ \bar{C}_3 m^{-1/6} \sqrt{\log m} L^4 T^{5/3} \lambda^{-5/3} (1 + \sqrt{T/\lambda})$$

Based on these notations, we state the following results from the previous works.

## D.1  Auxiliary Lemmas

**Lemma D.1** (Lemma 4.2 of [42]). *For any $t \in [T]$, $\mathbb{P}(\mathcal{E}_t^\sigma) \ge 1 - t^{-2}$.*

**Lemma D.2** (Lemma 4.3 of [42]). *Set $\eta = C(m\lambda + mLT)^{-1}$, then we have $\mathbb{P}(\mathcal{E}_t^\mu) \ge 1 - \delta$.*

**Lemma D.3** (Lemma B.9 of [42]). *For any time $t \in [T]$, $i \in [K]$ and any $\delta \in (0, 1)$, if the network width $m$ satisfies Condition 1, we have with probability at least $1 - \delta$, that $\sigma_{i,t} \le C_4 \sqrt{L}$, where $C_4$ is a positive constant.*

**Lemma D.4** (Lemma 4.4 of [42]). *For any $t \in [T]$, $i \in [K]$, we have $\mathbb{P}[\tilde{r}_{i,t} + \epsilon(m) > h(x_{i,t})|\mathcal{F}_t] \ge (4e\sqrt{\pi})^{-1}$.*

**Lemma D.5** (Lemma 4.8 of [42]). *Assume that the width of the neural network $m$ satisfies*

$$m \ge C\mathrm{max}\{\sqrt{\lambda}L^{-3/2}[log(TKL^2/\delta)]^{3/2}, T^6 K^6 L^6 log(TKL/\delta)\mathrm{max}\{\lambda_0^{-4}, 1\}\}$$

*and*

$m[log(m)]^{-3} \ge CTL^{12}\lambda^{-1} + CT^7\lambda^{-8}L^{18}(\lambda + LT)^6 + CL^{21}T^7\lambda^{-7}(1 + \sqrt{T/\lambda})^6$, where $C$ is a positive absolute constant.

*Then $\eta = C_5(m\lambda + mLT)^{-1}$. With high probability $1 - \delta$, for every $i \in [K]$ we have*

$$\sum_{i=1}^T min\{\sigma_{a_t,t}, 1\} \le \sqrt{2\lambda T(\widetilde{d}log(1 + TK) + 1)} + C6T^{13/6}\sqrt{logm}m^{-1/6}\lambda^{-2/3}L^{9/2},$$

*where $C_5, C_6$ are absolute constants.*

## D.2  Proof for Theorem 4.2

We now are ready to derive the regret bound for NeuralEI algorithm. First, we need the following results to lower bound and upper bound the EI acquisition function in the setting of neural contextual bandits. We remark that the following Lemma D.6 is an adaptation of Lemma C.4 (in linear bandits) to neural contextual bandits. The following Lemma D.7 is our novel result. Both Lemma D.6 and Lemma D.7 provide the lower bounds and the upper bounds for the EI acquisition function, but follow the different ways.

**Lemma D.6.** *Assume that the event $\mathcal{E}_t^\sigma$ holds. Set $I_{i,t} = max\{0, \tilde{f}_{i,t} - f_t^+\}$. Then for every $i \in [K]$, we have*

$$I_{i,t} - c_t \nu \sigma_{i,t} \le \alpha_{i,t}^{EI} \le I_{i,t} + \nu(c_t + 1)\sigma_{i,t}.$$

*Proof.* If $\sigma_{i,t} = 0$ then $\alpha_{i,t}^{EI} = I_{i,t}$, which makes the result trivial. We now assume that $\sigma_{i,t} > 0$. Set $q = \frac{\tilde{f}_{i,t} - f_t^+}{\sigma_{i,t}}$ and $u = \frac{f(x_{i,t}; \theta_{t-1}) - f_t^+}{\sigma_{i,t}}$. Set $\tau(z) = z\Phi(z) + \phi(z)$. Then we have that

$$\alpha_{i,t}^{EI} = \nu \sigma_{i,t} \tau(\frac{u}{\nu}).$$

By Lemma D.1, we have that $|u - q| \le c_t \nu$ with probability $1 - t^{-2}$. As $\tau'(z) = \Phi(z) \in [0, 1]$, $\tau$ is non-decreasing and $\tau(z) \le 1 + z$ for $z > 0$. Hence,

$$
\begin{aligned}
\alpha_{i,t}^{EI} &\le \nu \sigma_{i,t} \tau \left( \frac{\max\{0, q\} + c_t \nu}{\nu} \right) \\
&\le \nu \sigma_{i,t} \left( \frac{\max\{0, q\} + c_t \nu}{\nu} + 1 \right) \\
&= I_{i,t} + (c_t \nu + \nu) \sigma_{i,t}
\end{aligned}
$$

If $I_{i,t} = 0$ then the lower bound is trivial as $\alpha_{i,t}^{EI}$ is non-negative. Thus suppose $I_{i,t} > 0$. Since $\alpha_{i,t}^{EI} \ge 0$ and $\tau(z) \ge 0$ for all $z$, and $\tau(z) = z + \tau(-z) \ge z$. Therefore,

$$
\begin{aligned}
\alpha_{i,t}^{EI} &\ge \nu \sigma_{i,t} \tau \left( \frac{q - c_t \nu}{\nu} \right) \\
&\ge \nu \sigma_{i,t} \left( \frac{q - c_t \nu}{\nu} \right) \\
&= I_{i,t} - c_t \nu \sigma_{i,t}
\end{aligned}
$$

$\square$

**Lemma D.7.** *Assume that the event $\mathcal{E}_t^{\mu}$ holds. Set $J_{i,t} = \max\{0, h(x_{i,t}) - f_t^+\}$. Then for every $i \in [K]$, we have*

$$
J_{i,t} - \nu \sigma_{i,t} - \epsilon(m) \le \alpha_{i,t}^{EI} \le J_{i,t} + \nu \sigma_{i,t} + \epsilon(m).
$$

*Proof.* If $\sigma_{i,t} = 0$ then $\alpha_{i,t}^{EI} = I_{i,t}$, which makes the result trivial. We now assume that $\sigma_{i,t} > 0$. Set $q = \frac{h(x_{i,t}) - f_t^+}{\sigma_{i,t}}$ and $u = \frac{f(x_{i,t}; \theta_{t-1}) - f_t^+}{\sigma_{i,t}}$. Set $\tau(z) = z\Phi(z) + \phi(z)$. Then we have that

$$
\alpha_{i,t}^{EI} = \nu \sigma_{i,t} \tau \left( \frac{u}{\nu} \right).
$$

By Lemma D.2, we have that $|u - q| \le \nu + \frac{\epsilon(m)}{\sigma_{i,t}}$. As $\tau'(z) = \Phi(z) \in [0, 1]$, $\tau$ is non-decreasing and $\tau(z) \le 1 + z$ for $z > 0$. Hence,

$$
\begin{aligned}
\alpha_{i,t}^{EI} &\le \nu \sigma_{i,t} \tau \left( \frac{\max\{0, q\} + \nu + \frac{\epsilon(m)}{\sigma_{i,t}}}{\nu} \right) \\
&\le \nu \sigma_{i,t} \left( \frac{\max\{0, q\} + \nu + \frac{\epsilon(m)}{\sigma_{i,t}}}{\nu} + 1 \right) \\
&= J_{i,t} + \nu \sigma_{i,t} + \epsilon(m)
\end{aligned}
$$

If $I_{i,t} = 0$ then the lower bound is trivial as $\alpha_{i,t}^{EI}$ is non-negative. Thus suppose $I_{i,t} > 0$. Since $\alpha_{i,t}^{EI} \ge 0$ and $\tau(z) \ge 0$ for all $z$, and $\tau(z) = z + \tau(-z) \ge z$. Therefore,

$$
\begin{aligned}
\alpha_{i,t}^{EI} &\ge \nu \sigma_{i,t} \tau \left( \frac{q - \nu - \frac{\epsilon(m)}{\sigma_{i,t}}}{\nu} \right) \\
&\ge \nu \sigma_{i,t} \left( \frac{q - \nu - \frac{\epsilon(m)}{\sigma_{i,t}}}{\nu} \right) \\
&= J_{i,t} - \nu \sigma_{i,t} - \epsilon(m)
\end{aligned}
$$

$\square$

Next, we will use above results to bound $h(x_{a^*(t),t}) - h(x_{a(t),t})$. By NeuralEI algorithm, we will need to bound $h(x_{a^*(t),t}) - h(x_{a(t),t})$ thought different ways depending on the definition of $a(t)$.

### D.2.1 Considering the case $a(t) = \bar{a}(t)$

**Lemma D.8.** *Assume that the event $\mathcal{E}_t^\sigma$ holds. Then for every $i \in [K]$, we have*

$$\tilde{f}_{i,t} - f_t^+ \leq \frac{\tau(c_t)}{\tau(-c_t)}(2c_t + 1)\nu\sigma_{a(t),t}.$$

*Proof.* We provide the proof in Section E □

**Lemma D.9.** *Assume that the event $\mathcal{E}_t^\mu$ holds. Then for every $i \in [K] : \nu\sigma_{i,t} > \epsilon(m)$, we have*

$$h(x_{i,t}) - f_t^+ \leq (\frac{\tau(2)}{\tau(-2)} + 1)(\nu\sigma_{a(t),t} + \epsilon(m)).$$

*Proof.* We provide the proof in Section E □

**Lemma D.10.** *Assume that the event $\mathcal{E}_t^\mu$ holds. Pick a $\delta \in (0,1)$. Then with probability $1 - \delta$ we have*

$$f_t^+ - h(x_{a(t),t}) \leq (\sqrt{2log(C_1 C_0^{-1}\sqrt{L}\nu t^\beta)} + 1)\nu\sigma_{a(t),t} + \epsilon(m).$$

*Proof.* We have

$$
\begin{aligned}
\alpha_{a(t),t}^{EI} &= (f(x_{a(t),t}; \theta_{t-1}) - f_t^+)\Phi(\frac{f(x_{a(t),t}; \theta_{t-1}) - f_t^+}{\nu\sigma_{a(t),t}}) + \nu\sigma_{a(t),t}\phi(\frac{f(x_{a(t),t}; \theta_{t-1}) - f_t^+}{\nu\sigma_{a(t),t}}) \\
&\leq \nu\sigma_{a(t),t}\phi(\frac{f(x_{a(t),t}; \theta_{t-1}) - f_t^+}{\nu\sigma_{a(t),t}}) \\
&= \nu\sigma_{a(t),t}\frac{1}{2\sqrt{\pi}}\exp(-\frac{1}{2}(\frac{f(x_{a(t),t}; \theta_{t-1}) - f_t^+}{\nu\sigma_{a(t),t}})^2)
\end{aligned}
$$

where the first inequality holds due to the definition of action $a(t)$. The second equality holds due to the definition of $\alpha_{a(t),t}^{EI}$. The second inequality comes from the fact that $f(x_{a(t),t}; \theta_{t-1}) \leq f_t^+$. The third equality holds due to the definition of function $\phi(.)$.

From the last inequality, we obtain

$$|f(x_{a(t),t}; \theta_{t-1}) - f_t^+| \leq \sqrt{2log(\frac{\nu\sigma_{a(t),t}}{\alpha_{a(t),t}^{EI}})}\nu\sigma_{a(t),t}.$$

By using the condition that $\alpha_{a(t),t}^{EI} \geq \frac{C_0}{t^\beta}$ and the fact that $\sigma_{a(t),t} \leq C_1\sqrt{L}$ with probability $1 - \delta$, we have

$$|f(x_{a(t),t}; \theta_{t-1}) - f_t^+| \leq \sqrt{2log(C_1 C_0^{-1}\sqrt{L}\nu t^\beta)}\nu\sigma_{a(t),t} \tag{8}$$

with probability $1 - \delta$.

Finally, with probability $1 - \delta$, we have

$$
\begin{aligned}
f_t^+ - h(x_{a(t),t}) &= f_t^+ - f(x_{a(t),t}; \theta_{t-1}) + f(x_{a(t),t}; \theta_{t-1}) - h(x_{a(t),t}) \\
&\leq \sqrt{2log(C_1 C_0^{-1}\sqrt{L}\nu t^\beta)}\nu\sigma_{a(t),t} + \nu\sigma_{a(t),t} + \epsilon(m) \\
&= (\sqrt{2log(C_1 C_0^{-1}\sqrt{L}\nu t^\beta)} + 1)\nu\sigma_{a(t),t} + \epsilon(m),
\end{aligned}
$$

where in the first inequality, we use the inequality Eq(8) and Lemma D.2 □

**Lemma D.11.** *We assume that both $\mathcal{E}_t^\sigma$ and $\mathcal{E}_t^\mu$ hold. Then,*

- *if $\sigma_{a^*(t),t} > \epsilon(m)$, then*

$$h(x_{a^*(t),t}) - h(x_{a(t),t}) \le (\frac{\tau(2)}{\tau(-2)} + 1)(\nu\sigma_{a(t),t} + \epsilon(m)) + (\sqrt{2log(C_1 C_0^{-1}\sqrt{L}\nu t^\beta)} + 1)\nu\sigma_{a(t),t} + \epsilon(m)$$

- *if $\sigma_{a^*(t),t} \le \epsilon(m)$, then*

$$h(x_{a^*(t),t}) - h(x_{a(t),t}) \le \frac{\tau(c_t)}{\tau(-c_t)}\nu(2c_t + 1)\sigma_{a(t),t} + (\sqrt{2log(C_1 C_0^{-1}\sqrt{L}\nu t^\beta)} + 1)\nu\sigma_{a(t),t} + (\nu + c_t\nu + 1)\epsilon(m)$$

*Proof.* We consider two cases:

- if $\sigma_{a^*(t),t} > \epsilon(m)$. Then, by Lemma D.9, we have

$$h(x_{a^*(t),t}) - f_t^+ \le (\frac{\tau(2)}{\tau(-2)} + 1)(\nu\sigma_{a(t),t} + \epsilon(m)).$$

Thus,

$$
\begin{aligned}
h(x_{a^*(t),t}) - h(x_{a(t),t}) &= [h(x_{a^*(t),t}) - f_t^+] + [f_t^+ - h(x_{a(t),t})] \\
&\le (\frac{\tau(2)}{\tau(-2)} + 1)(\nu\sigma_{a(t),t} + \epsilon(m)) + (\sqrt{2log(C_1 C_0^{-1}\sqrt{L}\nu t^\beta)} + 1)\nu\sigma_{a(t),t} + \epsilon(m)
\end{aligned}
$$

- if $\sigma_{a^*(t),t} \le \epsilon(m)$. We have that $|h(x_{a^*(t),t}) - \tilde{f}_{a^*(t),t}| \le |h(x_{a^*(t),t}) - f(x_{a^*(t),t}; \theta_t)| + |f(x_{a^*(t),t}; \theta_t) - \tilde{f}_{a^*(t),t}| \le \epsilon(m) + (1 + c_t)\nu\sigma_{a^*(t),t} \le (\nu + c_t\nu + 1)\epsilon(m)$. Combining this result with Lemma D.8 and D.9, we have

$$
\begin{aligned}
h(x_{a^*(t),t}) - h(x_{a(t),t}) &= [h(x_{a^*(t),t}) - \tilde{f}_{a^*(t),t}] + [\tilde{f}_{a^*(t),t} - f_t^+] + [f_t^+ - h(x_{a(t),t})] \\
&\le (\nu + c_t\nu + 1)\epsilon(m) + \frac{\tau(c_t)}{\tau(-c_t)}\nu(2c_t + 1)\sigma_{a(t),t} + \\
&\quad + (\sqrt{2log(C_1 C_0^{-1}\sqrt{L}\nu t^\beta)} + 1)\nu\sigma_{a(t),t} + \epsilon(m) \\
&= \frac{\tau(c_t)}{\tau(-c_t)}\nu(2c_t + 1)\sigma_{a(t),t} + (\sqrt{2log(C_1 C_0^{-1}\sqrt{L}\nu t^\beta)} + 1)\nu\sigma_{a(t),t} + \\
&\quad (\nu + c_t\nu + 1)\epsilon(m),
\end{aligned}
$$

where in the above inequality, we apply Lemma D.8 for the arm $i_0$.

$\square$

### D.2.2 Considering the case $a(t) = \mathbf{argmax}_{i\in[K]}\{f(x_{i,t}; \theta_{t-1})\}$

**Lemma D.12.** *We assume that both $\mathcal{E}_t^\sigma$ holds. Then for $i \in [K]$ we have*

$$\tilde{f}_{i,t} - f_t^+ \le \frac{\tau(c_t)}{\tau(-c_t)}\frac{C_0}{t^\beta}.$$

*Proof.* The proof is similar to that of Lemma D.8 with a notice that $\alpha_{a(t),t}^{EI} \le \frac{C_0}{t^\beta}$ $\square$

**Lemma D.13.** *Assume that the event $\mathcal{E}_t^\mu$ holds. Then for every $i \in [K] : \nu\sigma_{i,t} > \epsilon(m)$, we have*

$$h(x_{i,t}) - f_t^+ \le (\frac{\tau(2)}{\tau(-2)} + 1)\frac{C_0}{t^\beta}.$$

*Proof.* The proof is similar to that of Lemma D.9 with a notice that $\alpha_{a(t),t}^{EI} \le \frac{C_0}{t^\beta}$ $\square$

**Lemma D.14.** *We assume that $\mathcal{E}_t^\mu$ holds. Then we have*

$$f_t^+ - h(x_{a(t),t}) \leq \nu\sigma_{a(t),t} + \epsilon(m).$$

*Proof.* By definition, $a(t) = \operatorname{argmax}_{i\in[K]}\{f(x_{i,t};\theta_{t-1})\}$. Hence $f_t^+ = f(x_{a(t),t};\theta_{t-1})$. By Lemma D.2, we have $f_t^+ - h(x_{a(t),t}) = f(x_{a(t),t};\theta_{t-1}) - h(x_{a(t),t}) \leq \nu\sigma_{a(t),t} + \epsilon(m)$. The lemma holds. $\square$

By combining the bounds of $h(x_{a^*(t),t}) - h(x_{a(t),t})$ for two cases of $a(t)$, we obtain an unique bound for $h(x_{a^*(t),t}) - h(x_{a(t),t})$ as follows:

**Lemma D.15.** *We assume that both $\mathcal{E}_t^\sigma$ and $\mathcal{E}_t^\mu$ hold. Then,*

- *if $\sigma_{a^*(t),t} > \epsilon(m)$, then,*

$$h(x_{a^*(t),t}) - h(x_{a(t),t}) \leq \left(\frac{\tau(2)}{\tau(-2)} + 1\right)\frac{C_0}{t^\beta} + \nu\sigma_{a(t),t} + \epsilon(m).$$

- *if $\sigma_{a^*(t),t} \leq \epsilon(m)$, then*

$$h(x_{a^*(t),t}) - h(x_{a(t),t}) \leq (\nu + c_t\nu + 2)\epsilon(m) + \frac{\tau(c_t)}{\tau(-c_t)}\frac{C_0}{t^\beta} + \nu\sigma_{a(t),t}.$$

*Proof.* There are two cases:

- if $\sigma_{a^*(t),t} > \epsilon(m)$, then, by Lemma D.13 for action $a^*(t)$, we have

$$h(x_{a^*(t),t}) - f_t^+ \leq \left(\frac{\tau(2)}{\tau(-2)} + 1\right)\frac{C_0}{t^\beta}.$$

  Combining with Lemma D.14, we have

$$\begin{aligned}
h(x_{a^*(t),t}) - h(x_{a(t),t}) &= [h(x_{a^*(t),t}) - f_t^+] + [f_t^+ - h(x_{a(t),t})] \\
&\leq \left(\frac{\tau(2)}{\tau(-2)} + 1\right)\frac{C_0}{t^\beta} + \nu\sigma_{a(t),t} + \epsilon(m),
\end{aligned}$$

- if $\sigma_{a^*(t),t} \leq \epsilon(m)$, then $|h(x_{a^*(t),t}) - \tilde{f}_{a^*(t),t}| \leq |h(x_{a^*(t),t}) - f(x_{a^*(t),t};\theta_t)| + |f(x_{a^*(t),t};\theta_t) - \tilde{f}_{a^*(t),t}| \leq \epsilon(m) + (1+c_t)\nu\sigma_{a^*(t),t} \leq (\nu+c_t\nu+1)\epsilon(m)$. Thus, we have Combining with Lemma D.12 and Lemma D.14, we have

$$\begin{aligned}
h(x_{a^*(t),t}) - h(x_{a(t),t}) &= [h(x_{a^*(t),t}) - \tilde{f}_{a^*(t),t}] + [\tilde{f}_{a^*(t),t} - f_t^+] + [f_t^+ - h(x_{a(t),t})] \\
&\leq (\nu + c_t\nu + 2)\epsilon(m) + \frac{\tau(c_t)}{\tau(-c_t)}\frac{C_0}{t^\beta} + \nu\sigma_{a(t),t}
\end{aligned}$$

.

$\square$

**Lemma D.16.** *We assume that both $\mathcal{E}_t^\sigma$ and $\mathcal{E}_t^\mu$ hold. Then, there exist constants $C_1, C_2, C_3$ such that with probability $1 - \delta$, we have*

$$h(x_{a^*(t),t}) - h(x_{a(t),t}) \leq C_1(2c_t+1)\nu\sigma_{a(t),t} + \left(\sqrt{2log(C_1C_0^{-1}\sqrt{L}\nu t^\beta)}+1)\right)\nu\sigma_{a(t),t} + C_2\frac{C_0}{t^\beta} + (\nu+c_t\nu+C_3)\epsilon(m).$$

*Proof.* This upper bound is obtained by combining the upper bounds of $h(x_{a^*(t),t}) - h(x_{a(t),t})$ from Lemma D.11 and Lemma D.15. We note $\frac{\tau(c_t)}{\tau(-c_t)}$ is bounded by a constant. $\square$

Set $r_t = h(x_{a^*(t),t}) - h(x_{a(t),t})$. We can re-write the above lemma as follows:

**Lemma D.17.** *With probability $1 - \delta$, we have*

$$
\begin{aligned}
\mathbb{E}[r_t|\mathcal{F}_t, \mathcal{E}_t^\mu] &\leq (C_1(2c_t + 1) + (\sqrt{2log(C_1C_0^{-1}\sqrt{L}\nu t^\beta)} + 1)C_1\sqrt{L})\nu\mathbb{E}[min\{\sigma_{a(t),t}, 1\}|\mathcal{F}_t, \mathcal{E}_t^\mu] \\
&\quad + C_2\frac{C_0}{t^\beta} + (\nu + c_t\nu + C_3)\epsilon(m)
\end{aligned}
$$

*Proof.* We have

$$
\begin{aligned}
\mathbb{E}[r_t|\mathcal{F}_t, \mathcal{E}_t^\mu] &= \mathbb{E}[h(x_{a^*(t),t}) - h(x_{a(t),t})|\mathcal{F}_t, \mathcal{E}_t^\mu, \mathcal{E}_t^\sigma]\mathbb{P}(E_t^\sigma) + \\
&\quad + \mathbb{E}[h(x_{a^*(t),t}) - h(x_{a(t),t})|\mathcal{F}_t, \mathcal{E}_t^\mu, \overline{\mathcal{E}}_t^\sigma]\mathbb{P}(\overline{\mathcal{E}}_t^\sigma) \\
&\leq (C_1(2c_t + 1) + (\sqrt{2\log(C_1C_0^{-1}\sqrt{L}\nu t^\beta)} + 1))\nu\mathbb{E}[\sigma_{a(t),t}|\mathcal{F}_t, \mathcal{E}_t^\mu] + \\
&\quad + C_2\frac{C_0}{t^\beta} + (\nu + c_t\nu + C_3)\epsilon(m) + \frac{2}{t^2},
\end{aligned}
$$

where the first term is bounded by Lemma D.17 and the second term is bounded due to the facts that $h(x_{a^*(t),t}) - h(x_{a(t),t}) \leq |h(x_{a^*(t),t})| + |h(x_{a(t),t})| \leq 2$ and $\mathbb{P}(\overline{\mathcal{E}}_t^\sigma) \leq \frac{1}{t^2}$.

Further, due to $|h(x)| \leq 1$, we obtain an upper bound for $\mathbb{E}[r_t|\mathcal{F}_t, \mathcal{E}_t^\mu]$ as

$$
\begin{aligned}
\mathbb{E}[r_t|\mathcal{F}_t, \mathcal{E}_t^\mu] &\leq \min\{((C_1(2c_t + 1) + (\sqrt{2\log(C_1C_0^{-1}\sqrt{L}\nu t^\beta)} + 1))\nu)\mathbb{E}[\sigma_{a(t),t}|\mathcal{F}_t, \mathcal{E}_t^\mu], 2\} \\
&\quad + C_2\frac{C_0}{t^\beta} + (\nu + c_t\nu + C_3)\epsilon(m) + \frac{2}{t^2}
\end{aligned}
$$

On the other hand, we have $\nu \geq 1$ by definition, $2c_t + 1 = 2\sqrt{4\log t + 2\log K} + 1 \geq 2$ with $t \geq 2$. The constant $C_1$ is chosen to be greater than 1. Hence,

$$
\begin{aligned}
\mathbb{E}[r_t|\mathcal{F}_t, \mathcal{E}_t^\mu] &\leq ((C_1(2c_t + 1) + (\sqrt{2\log(C_1C_0^{-1}\sqrt{L}\nu t^\beta)} + 1))\nu)\min\{\mathbb{E}[\sigma_{a(t),t}|\mathcal{F}_t, \mathcal{E}_t^\mu], 1\} \\
&\quad + C_2\frac{C_0}{t^\beta} + (\nu + c_t\nu + C_3)\epsilon(m) + \frac{2}{t^2} \\
&\leq ((C_1(2c_t + 1) + (\sqrt{2\log(C_1C_0^{-1}\sqrt{L}\nu t^\beta)} + 1))\nu)C_4\sqrt{L})\mathbb{E}[\min\{\sigma_{a(t),t}, 1\}|\mathcal{F}_t, \mathcal{E}_t^\mu] \\
&\quad + C_2\frac{C_0}{t^\beta} + (\nu + c_t\nu + C_3)\epsilon(m) + \frac{2}{t^2},
\end{aligned}
$$

where we use Lemma D.3 with probability $1 - \delta$. $\qquad\square$

We now are ready to prove Theorem 4.2.

**Theorem D.18.** *If the network width $m$ satisfies:*

$$
m \geq poly(\gamma, T, K, L, log(1/\delta)),
$$

*then with probability at least $1 - \delta$, the regret of Algorithm 2 is bounded as*

$$
R_T \leq C_2(1 + c_T)\nu\sqrt{2\lambda L(\widetilde{d}log(1 + TK) + 1)T} + (4 + C_3(1 + c_T)\nu L)\sqrt{2log(3/\delta)T} + 5.
$$

*Proof.* By Lemma D.2, the event $\mathcal{E}_t^\mu$ holds for all $1 \leq t \leq T$ with probability at least $1 - \delta$. Hence, with probability $1 - \delta$ we have

$$
\begin{aligned}
\overline{R}(T) &= \sum_{i=1}^T h(x_{a^*(t),t}) - h(x_{a(t),t})\mathbf{1}(\mathcal{E}_t^\mu) \\
&\leq ((C_1(2c_t + 1) + (\sqrt{2\log(C_1C_0^{-1}\sqrt{L}\nu t^\beta)} + 1))\nu)C_4\sqrt{L}\sum_{i=1}^T \mathbb{E}[\min\{\sigma_{a(t),t}, 1\}|\mathcal{F}_t, \mathcal{E}_t^\mu] \\
&\quad + \sum_{i=1}^T C_2\frac{C_0}{t^\beta} + (\nu + c_t\nu + C_3)\epsilon(m) + \frac{2}{t^2}
\end{aligned}
$$

$$\leq \quad MN + C_2 C_0 \sum_{i=1}^{T} \frac{1}{t^\beta} + (\nu + c_t\nu + C_3)\epsilon(m)T + \frac{\pi^2}{3},$$

where we set $M = ((C_1(2c_t + 1) + (\sqrt{2\log(C_1 C_0^{-1}\sqrt{L}\nu t^\beta)} + 1))\nu)C_4\sqrt{L}$ and $N = (\sqrt{2\lambda T(\tilde{d}\log(1 + TK) + 1)} + C2T^{13/6}\sqrt{\log m}m^{-1/6}\lambda^{-2/3}L^{9/2})$ which is obtained by Lemma D.5.

With $\beta \geq 2$, $\sum_{i=1}^{T} \frac{1}{t^\beta} < \frac{\pi^2}{6}$. We choose $C_0 \in [\sqrt{\tilde{d}}, \tilde{d}]$ to eliminate the term $\tilde{d}$ in $\sqrt{2\log(C_1 C_0^{-1}\sqrt{L}\nu T^\beta)}$.

By choosing $m$ similar as in [42] such that $C_3(2c_T + 1)C2T^{13/6}\sqrt{\log m}m^{-1/6}\lambda^{-2/3}L^5) \leq \frac{1}{3}$, $(\nu + c_t\nu + C_3)\epsilon(m)T \leq \frac{1}{3}$, we obtain

$$\overline{R}(T) \leq \mathcal{O}(\tilde{d}\sqrt{\beta\log(1 + TK)\log(T)T})$$

$\square$

# E   Additional Lemmas

### E.0.1   Proof of Lemma D.8

*Proof.* If $\sigma_{i,t} = 0$ then by definition of $\alpha_{i,t}^{EI}$, we have $\alpha_{i,t}^{EI} = I_{i,t}$. We have

$$\begin{aligned}
I_{i,t} &= \alpha_{i,t}^{EI} \\
&\leq \alpha_{a(t),t}^{EI} \\
&\leq I_{a(t),t} + \nu(c_t + 1)\sigma_{a(t),t},
\end{aligned}$$

where in the first inequality, we use the definition $\alpha_{a(t),t}^{EI} = \max_{i\in[K]}\alpha_{i,t}^{EI}$. In the second inequality, we use Lemma D.1. Thus, the lemma holds with probability $1 - t^{-2}$.

We now consider $\sigma_{i,t} > 0$. If $\tilde{f}_{i,t} < f_t^+$ then the lemma will be trivial. We now consider $\tilde{f}_{i,t} > f_t^+$. Following the derivation of the acquisition function $\alpha_{i,t}^{EI}$, we have $\alpha_{i,t}^{EI} = \nu\sigma_{i,t}\tau(\frac{f(x_{i,t};\theta_{t-1}) - f_t^+}{\nu\sigma_{i,t}})$. Further, we also have $\frac{f(x_{i,t};\theta_{t-1}) - f_t^+}{\nu\sigma_{i,t}} \geq \frac{-c_t\nu}{\nu}$. It is because $f(x_{i,t};\theta_{t-1}) - f(x_{i,t}) \geq -c_t\nu\sigma_{i,t}$ and we are considering the case when $\tilde{f}_{i,t} > f_t^+$. Therefore, $\alpha_{i,t}^{EI} \geq \nu\tau(-c_t)\sigma_{i,t}$.

Now, we combine the fact that $\alpha_{i,t}^{EI} \geq \nu\tau(-c_t)\sigma_{i,t}$ with the fact that $\alpha_{i,t}^{EI} \geq I_{i,t} - c_t\nu\sigma_{i,t}$ which is proven in Lemma D.6 to obtain the following inequality:

$$I_{i,t} \leq \frac{\tau(c_t)}{\tau(-c_t)}\alpha_{i,t}^{EI} \tag{9}$$

This inequality Eq(9) holds. Finally, we achieve

$$\begin{aligned}
\tilde{f}_{i,t} - f_t^+ &\leq I_{i,t} \\
&\leq \frac{\tau(c_t)}{\tau(-c_t)}\alpha_{i,t}^{EI} \\
&\leq \frac{\tau(c_t)}{\tau(-c_t)}\alpha_{a(t),t}^{EI} \\
&\leq \frac{\tau(c_t)}{\tau(-c_t)}(\max\{0, \tilde{f}_{a(t),t} - f_t^+\} + \nu(c_t + 1)\sigma_{a(t),t} \\
&\leq \frac{\tau(c_t)}{\tau(-c_t)}(\max\{0, f(x_{a(t),t};\theta_{t-1}) + c_t\nu\sigma_{a(t),t} - f_t^+\} + \nu(c_t + 1)\sigma_{a(t),t} \\
&\leq \frac{\tau(c_t)}{\tau(-c_t)}(\max\{0, c_t\nu\sigma_{a(t),t}\} + \nu(c_t + 1)\sigma_{a(t),t}
\end{aligned}$$

$$= \frac{\tau(c_t)}{\tau(-c_t)}(2c_t\nu + \nu)\sigma_{a(t),t}$$

where the first inequality holds by the definition of the function $I_t$. The second one comes from Eq(9). The third one holds by the property of the chosen point $a(t) = \text{argmax}_{i\in[K]}\alpha_{i,t}^{EI}$. The final inequality hold due to Lemma D.6. $\qquad\square$

### E.0.2 Proof of Lemma D.9

*Proof.* If $\sigma_{i,t} = 0$ then by definition of $\alpha_{i,t}^{EI}$, we have $\alpha_{i,t}^{EI} = I_{i,t}$. We have

$$\begin{aligned}
I_{i,t} &= \alpha_{i,t}^{EI} \\
&\leq \alpha_{a(t),t}^{EI} \\
&\leq I_{a(t),t} + \nu(c_t+1)\sigma_{a(t),t},
\end{aligned}$$

where in the first inequality, we use the definition $\alpha_{a(t),t}^{EI} = \max_{i\in[K]}\alpha_{i,t}^{EI}$. In the second inequality, we use Lemma D.2. Thus, the lemma holds with probability $1 - t^{-2}$.

We now consider $\sigma_{i,t} > 0$. If $h(x_{i,t}) < f_t^+$ then the lemma will be trivial. We now consider $h(x_{i,t}) > f_t^+$. Following the derivation of the acquisition function $\alpha_{i,t}^{EI}$, we have $\alpha_{i,t}^{EI} = \nu\sigma_{i,t}\tau(\frac{f(x_{i,t};\theta_{t-1})-f_t^+}{\nu\sigma_{i,t}})$. Further, we also have $\frac{f(x_{i,t};\theta_{t-1})-f_t^+}{\nu\sigma_{i,t}} \geq -1 - \frac{\epsilon(m)}{\nu\sigma_{i,t}}$. It is because $f(x_{i,t};\theta_{t-1}) - h(x_{i,t}) \geq -\nu\sigma_{i,t} - \epsilon(m)$ and we are considering the case when $h(x_{i,t}) > f_t^+$. Therefore, $\alpha_{i,t}^{EI} \geq \nu\sigma_{i,t}\tau(-1 - \frac{\epsilon(m)}{\nu\sigma_{i,t}})$.

Now, we combine the fact that $\alpha_{i,t}^{EI} \geq \nu\sigma_{i,t}\tau(-1-\frac{\epsilon(m)}{\nu\sigma_{i,t}})$ with the fact that $\alpha_{i,t}^{EI} \geq J_{i,t} - \nu\sigma_{i,t} - \epsilon(m)$ which is proven in Lemma D.6 to obtain the following inequality:

$$I_{i,t} \leq \frac{\tau(1 + \frac{\epsilon(m)}{\nu\sigma_{i,t}})}{\tau(-1 - \frac{\epsilon(m)}{\nu\sigma_{i,t}})}\alpha_{i,t}^{EI}.$$

Using the assumption that $\nu\sigma_{i,t} > \epsilon(m)$, we get

$$I_{i,t} \leq \frac{\tau(2)}{\tau(-2)}\alpha_{i,t}^{EI}$$

This inequality Eq(10) holds. Finally, we achieve

$$\begin{aligned}
\tilde{f}_{i,t} - f_t^+ &\leq I_{i,t} \\
&\leq \frac{\tau(2)}{\tau(-2)}\alpha_{i,t}^{EI} \\
&\leq \frac{\tau(2)}{\tau(-2)}\alpha_{a(t),t}^{EI} \\
&\leq \frac{\tau(2)}{\tau(-2)}(\max\{0, h(x_{a(t),t}) - f_t^+\} + \nu\sigma_{a(t),t} + \epsilon(m)) \\
&\leq \frac{\tau(2)}{\tau(-2)}(\max\{0, f(x_{a(t),t};\theta_{t-1}) + \nu\sigma_{a(t),t} + \epsilon(m) - f_t^+\} + \nu\sigma_{a(t),t} + \epsilon(m)) \\
&\leq \frac{\tau(2)}{\tau(-2)}(\max\{0, \nu\sigma_{a(t),t} + \epsilon(m)\} + \nu\sigma_{a(t),t} + \epsilon(m)) \\
&= (\frac{\tau(2)}{\tau(-2)} + 1)(\nu\sigma_{a(t),t} + \epsilon(m))
\end{aligned}$$

where the first inequality holds by the definition of the function $I_t$. The second one comes from Eq(10). The third one holds by the property of the chosen point $a(t) = \text{argmax}_{i\in[K]}\alpha_{i,t}^{EI}$. The final inequality hold due to Lemma D.6. $\qquad\square$