# OpenReview forum: "Expected Improvement for Contextual Bandits"
_NeurIPS.cc/2022/Conference — NeurIPS 2022 Accept_

### Official Review · Reviewer_KPAe · 2022-07-08

**Rating:** 6
**Confidence:** 3
**Soundness:** 3 good
**Presentation:** 3 good
**Contribution:** 2 fair

**Summary:**

The expected improvement (EI) is a classic approach to select the action in Bayesian optimization (BO). This paper extends EI to the contextual bandit setting, including the linear setting and non-linear setting. Based on the fact, the analysis of EI is not well studied, this paper provides the regret analysis of EI in linear contextual bandit and neural contextual bandit. Moreover, authors show good empirical results proposed algorithm.

enecccdejhvddktidjggvrltrghcivbtjdnlekdkejtc


**Questions:**

(1) LinEI and NeuralEI, look like the expected version of TS, adding expectation over the sampled reward. Please author convince me of the significant difference from TS.

(2) Please provide more details about the challenges in the analysis of LinEI and NeuralEI over LinTS and NeuralTS.



**Limitations:**

.

**Strengths And Weaknesses:**



Weakness:

(1) The overall proposed two algorithms, LinEI and NeuralEI,  look very similar to TS to me. Take the linEI as an example. In a round, $r_t^+$ is the same for every arm. So the selection criterion can be considered as $E_{\mu \sim N}x^\top \mu $, which is the expected version of LinTS. Just add the expectation to the TS. Theoretically, we only need to take the expectation over the normal distribution based on TS. So, I doubt the novelty of the proposed algorithms and their analysis.

---

> ### Author Response · Authors · 2022-08-02
> **Please find responses to your questions below.**
>
> We thank you for your encouraging and constructive feedback.  We address your questions as follows:
>
> **LinEI and NeuralEI, look like the expected version of TS, adding expectation over the sampled reward. Please author convince me of the significant difference from TS**.
>
> We would like to emphasize that the EI here is not the same as the expected version of TS. To see this, consider the EI expression for contextual bandit case defined as   $E_{\mu \sim \mathcal N(\hat{\theta}_{t}, *)}[\text{max} (0, x^T\mu - r_t^+ )]$.
>
> Even though one may think that  $E_{\mu \sim N(\hat{\theta}_t,*)} x^T\mu = x^T\hat{\theta}_t$,  but the expectation operator cannot be taken inside the $\text{max}$ operator as it is a nonlinear function. It follows that the closed form of EI is more complicated (contains nonlinear components) as given in Eq(2) in the main paper. This closed form is similar to that of EI in the case of the multi-arm bandit [1].
> Further, an important aspect of TS is its randomization (before maximization) which allows it to perform the necessary explorations. To our knowledge, using an expectation in TS may disturb its exploration behavior, and it can behave as a greedy strategy that can bear a linear regret.
>
> **More details about the challenges in the analysis of LinEI and NeuralEI over LinTS and NeuralTS**.
>
> There are three major challenges (and therefore differences) in the analysis of LinEI over LinTS.
> 1. First, while the TS provides a simple form $x^T\mu$ to determine the arm to play at a round, the closed form of EI is more complicated due to non-linear components (see Equation (2) in our main paper). This makes it a challenge to bound the EI. In the other hand, also due to the simple form of TS, TS bears a concentration inequality
> $ |x_{a(t),t}^T \mu - r_t^+| \le \tilde{\mathcal O}(d s_{a(t),t})$,
> where $a(t)$ is the arm selected by LinTS, $s_{a(t),t}$ is the variance. This is stated in Lemma 1 in [2]. This concentration bound plays a central role in the analysis of LinTS. This inequality makes LinTS bear a regret bound which has an additional factor of $\sqrt{d}$ compared to the lower bound in this setting. We avoid this by deriving a novel concentration inequality for our LinEI as follows:
> $|x_{a(t),t}^T\theta^* - r_t^+| \le \tilde{\mathcal O}(\sqrt{d} s_{a(t),t})$, where $\theta^*$ is the unknown parameter of the true reward; $a(t)$ is the arm selected by the EI if the EI is higher than  $ \frac{C_0}{t^{\beta}}$  and by the greedy strategy otherwise. Please see  Lemmas C.6 and C.8 in our appendix for details. The concentration inequality with the order $\sqrt{d}$ allows the regret bound of LinEI to match the lower bound, and thus improve over the TS regret bound by a factor $\sqrt{d}$.
>
> 2. Second, we use a threshold in the form $\frac{C_0}{t^{\beta}}$ to switch between the EI and the greedy strategy. This technique is novel. The $\frac{C_0}{t^{\beta}}$ function has two important properties: (1) it is a decreasing function in time $t$; (2) the sum of $\frac{C_0}{t^{\beta}}$ for $t$ from 1 to infinity is bounded if $\beta \ge 2$. We exploit these properties in our regret bound analysis.
>
> 3. Finally, the regret analysis of TS is based on dividing the arms into two groups at any time: saturated and unsaturated arms (see [1]). The intuition is that the optimal arm is included in the group of unsaturated arms. The probability of playing saturated arms is small, and at every step an unsaturated arm will be played with some constant probability. Our LinEI does not follow this fashion. We directly bound the regret based on the selected arm no matter whether an arm is saturated or not.
>
> Our NeuralEI also does not follow the saturated-unsaturated fashion of NeuralTS although our analysis uses several results of NeuralTS as we mentioned in section D.1 in the appendix. NeuralEI extends techniques of LinEI to neural bandits with several challenges. These challenges come from the fact that there exists additionally a factor $\epsilon(m)$ in the concentration bound of the estimate of reward mean at Lemma D.3 in the appendix. Compared to LinEI, it is hard to analyze directly the regret $h(x_{a^*(t),t}) - h(x_{a(t),t})$, where $a(t)$ is the arm selected at round $t$ and $a^*(t)$ is the unknown best arm at round $t$. We divide the analysis into two cases: if $\sigma_{a^*(t),t} \le \epsilon(m)$, we can follow the technique as in LinEI. Otherwise, we use another way to bound the regret using a novel result as we mentioned in the main paper. Please see Lemma D.11 in the appendix for details.
>
> **References:**
>
> [1] Chao Qin, Diego Klabjan, and Daniel Russo. Improving the expected improvement algorithm. In Proceedings of the 31st International Conference on Neural Information Processing Systems, NIPS’17.
>
> [2]  Shipra Agrawal and Navin Goyal. Thompson sampling for contextual bandits with linear payoffs. Proceedings of the 30th International Conference on Machine Learning. 2013.

---

### Official Review · Reviewer_s7GF · 2022-07-09

**Rating:** 6
**Confidence:** 4
**Soundness:** 3 good
**Presentation:** 2 fair
**Contribution:** 3 good

**Summary:**

This paper applies the expected improvement (EI) principle to contextual bandits, which contrasts with the more popular approaches of upper-confidence bound (UCB) and Thompson sampling (TS).  For adversarially chosen contexts but realizable (conditional) mean rewards, EI is practically applicable with both linear and neural network predictors.  In the linear case the proposed technique essentially achieves information-theoretic lower bounds given sub-Gaussian residuals.  In the neural case the network tangent kernel can be used to characterize the regret matching analogous results for NeuralUCB and NeuralTS.

**Questions:**

For clarity state that $\xi_{a(t),t}$ is mean zero in line 71 (in some works, mean zero is an optional part of the definition of sub-gaussian).  For example, "... where $h$ is an unknown mean reward function ...".

I would appreciate some commentary around equation (1) and associated experiments in section 6.1.  For example, compared to equation (1), LinUCB will find the most optimistic value for an arm in an ellipsoid defined by a level set of the posterior, rather than integrating over it.  Thus, relative to LinUCB, LinEI could choose an arm whose maximum optimistic value is lower but for which there are "more possibilities" for improvement.  Try to provide similar comparative insight with LinTS.

theorem 3.2 and theorem 4.2 have typesetting issues.

The following combination of statements is confusing:
  * line 72: ... conditionally R sub-gaussian for some R >= 0 ...
  * line 230: We note that although our algorithm is a Bayesian approach, our regret bounds will hold irrespective of whether or not the actual reward distribution matches the Gaussian likelihood function ...
  * line 312: For example, we can extend our work to non-Gaussian distributions to model the unknown reward model parameters and derive regret bounds as long as concentration inequalities can be established.
    * I suspect what you want to say is: "we've modeled the conditional reward residuals as sub-gaussian; our regret bounds hold for all sub-gaussian residuals; and we want to extend our work to more general residuals such as sub-exponential or sub-gamma."  But to be honest, it's a bit unclear.  If my guess is correct, the issue is arguably marginal because: bounded rewards are guaranteed to have sub-gaussian residuals (hence your result is already broadly applicable), and you are already close to the information-theoretic lower bound in Theorem 3.2 (hence you have already squeezed most of the efficiency out of the approach).


**Limitations:**

This reviewer is satisfied.

**Update after author discussion period**:  Authors have addressed concerns, raising score.

**Strengths And Weaknesses:**

Strengths include the novelty of EI in this setting and the connection to NTK theory.

Weaknesses:
  * the lack of clear motivation for EI.
    * is it just "another way to do contextual bandits that nobody has done yet?"
      * if so this paper is likely to be ignored by practitioners.
    * is it statistically distinct from UCB/TS approaches (i.e., works better)?
      * in theory no, everybody is achieving the same regret bounds essentially.
      * in practice, maybe, but that's where the experiments are too limited for a strong conclusion.
      * one advantage of using older datasets is you can provide exhaustive comparative study as in https://arxiv.org/abs/1802.04064
    * is it computationally distinct from UCB/TS approaches (i.e., cheaper to compute or scales better to larger action sets)?
      * if so you could exhibit on a modern dataset such as amazon-3m from https://arxiv.org/abs/2102.07800

I believe this work would benefit from spending more time better differentiating EI before presenting to the general community, otherwise, it's potential impact will be limited.

---

> ### Author Response · Authors · 2022-08-02
> **Please find responses to your questions below**
>
> We thank you for your encouraging and constructive feedback. We address your questions as follows:
>
> **Is it statistically distinct from UCB/TS approaches (i.e., works better). In theory no, everybody is achieving the same regret bounds essentially.**
>
> We note that our proposed LinEI is statistically distinct and better than the TS-based approach for linear settings.
>
> - In the linear bandit setting, our proposed LinEI obtains a near-optimal regret $\tilde{\mathcal O} (d\sqrt{T})$ similar to LinUCB, however, the proposed LinEI improves upon LinTS by a factor of $\sqrt{d}$. [1] is the first work to prove theoretical bounds for LinTS and an alternative proof is given by [2]. Both papers show $\tilde{\mathcal O} (d^{3/2}\sqrt{T})$ regret bound, which is known as the best regret bound for LinTS. Very recently, a theoretical paper at NeurIPS 2021 [3] improved this regret bound by integrating a doubly robust estimator with TS. They obtain a regret bound $\tilde{\mathcal O} (\phi^{-2}\sqrt{T})$ where $\phi^2$ is the minimum eigenvalue of the covariance matrix of contexts. They show that $\phi^{-2} \ge d$ in general and $\phi^{-2} = d$ in only some cases. Thus, this regret bound is not near-optimal in general. In addition, this work requires additional significant computations and their setting is limited to independent contexts and requires the minimum eigenvalue of the covariance matrix of contexts to be bounded. In contrast, LinEI obtains a near-optimal regret in a general setting where we do not require the above conditions on contexts. This shows the benefit of EI for regret analysis compared to the TS-based approach.
> - We also compared the regret of LinEI with the regret of other works in our related work section.
>
> **The lack of motivation for EI**
>
> Besides the competitive regret and competitive experimental results, the EI also has a particular benefit in practice in comparison with UCB/TS. Both UCB and TS strategies can be considered over-explored.  In some situations, the exploration may costly or infeasible. For example, in medical decision-making, choosing a treatment that is not the estimated-best choice (pure exploration) for a specific patient may be unethical [4]. In this situation, our proposed EI chooses a safer treatment than UCB and TS. It is because EI only chooses a treatment with high possibilities for improvement over the estimated-best choice.
>
>
> **References**
>
> [1] Shipra Agrawal and Navin Goyal. Thompson sampling for contextual bandits with linear payoffs. In Sanjoy Dasgupta and David McAllester, editors, Proceedings of the 30th International Conference on Machine Learning, volume 28 of Proceedings of Machine Learning Research, pages 127–135, Atlanta, Georgia, USA, 17–19 Jun 2013.
>
> [2] Marc Abeille and Alessandro Lazaric. Linear Thompson Sampling Revisited. In Aarti Singh and Jerry Zhu, editors, Proceedings of the 20th International Conference on Artificial Intelligence and Statistics, volume 54 of Proceedings of Machine Learning Research, pages 176–184. PMLR, 20–22 Apr 2017.
>
> [3] Wonyoung Kim, Gi soo Kim, and Myunghee Cho Paik. Doubly robust thompson sampling for linear payoffs, 2021.
>
> [4] H. Bastani, M. Bayati, and K. Khosravi. Mostly exploration-free algorithms for contextual bandits. Management Science, 67(3):1329–1349, 2021.

---

> > ### Author Response · Authors · 2022-08-02
> > **Please find responses to your questions below**
> >
> > **Is it computationally distinct from UCB/TS approaches (i.e., cheaper to compute or scales better to larger action sets)?**
> >
> > Following your suggestion, we now compare our proposed LinEI against LinUCB and LinTS on the eurlex-4k dataset which is a large multi-label dataset [5]. It contains data with 5000 features and 3993 labels. It corresponds to a contextual bandit problem with 3993 actions and the dimension of context is $5000+3993 = 8993$ when mapped to our setting. This can be considered as a contextual bandit problem with large action space and with very high dimensions. Due to the heavy computation, we run algorithms on only 3000 datapoints from the test set. Our experimental results show that
> >
> > 1. LinEI scales better in high dimensions and in large action spaces (**please see Figure 5 in Section A.2 of our updated Supplementary**). This can be explained as the regret of LinEI is better than that of LinTS by a factor $\sqrt{d}$ while LinUCB is an over- exploration strategy.
> > 2. In terms of required computations, LinEI is significantly cheaper than LinUCB and LinTS (**please see Table 3 in Section A.2 of our updated Supplementary**). The average-per iteration computation times of LinUCB, LinTS and LinEI are 1055.5s, 1496.2s and 181.3s respectively. Interestingly, we find during our rebuttal that LinTS requires a multivariate Gaussian sampling step at each iteration. However, this is a high-dimensional multivariate Gaussian sampling problem [6]. It has computational costs and memory requirements that can rapidly become prohibitive in high dimensions. LinUCB requires to maximize quadratic forms at every round $1 \le t \le T$: $ \text{argmax } x_{t,i}^{T} \theta$, where $i \in [K]$, and $ \theta \in C_t$, where $C_t$ is the confident set at round $t$. We note that $\theta$ has the same dimension as the input $x_{t,i}$. This is an NP-hard problem as [1] has mentioned in the Related Work section.  When the input dimension is high, it is expensive to find a maximum with a limited computational budget. In contrast, the proposed LinEI can avoid this NP-hard problem as long as finding $\text{argmax } \alpha^{EI}_{i,t}$, where $ i \in [K]$ is solved. We can see that with a not very large value of $K$, this is no problem.
> > In contrast, our LinEI can avoid both the NP-hard problem and the high-dimensional Gaussian sampling problem. Thus, the computation cost of LinEI is significantly cheaper in high dimensions.
> >
> > Next, we additionally compare our proposed LinEI against LinUCB and LinTS on synthetic data in a range of dimensions d = 10, 100, and 1000. In this experiment, we use a contextual bandit problem with $K = 100$ actions. The context vectors are chosen uniformly at random from the unit ball. The reward function $r = x^T\theta^*$, where $\theta^*$ is generated \emph{a priori}.  We have provided our results in our **updated Supplementary material (see Section A.1)**.  Our results also show that (1)  LinEI scales to high dimensions better than LinUCB and LinTS, and (2)  the computation time of LinEI is significantly smaller compared to LinUCB and LinTS in high dimensions.
> >
> > **Try to provide similar comparative insight with LinTS.** Thanks for your suggestion! We now provided a comparison with LinTS in Section 3.1 of our revised main paper.  We restate it here:  "LinUCB looks for the most optimistic value for an arm in an ellipsoid defined by a level set of the posterior rather than integrating over it, and then chooses an arm that maximizes the optimistic value. LinTS generates a sample $\mu$ from the posterior distributions of reward and then chooses an arm that maximizes $x^T_{i,t}\mu$. Unlike LinUCB and LinTS, LinEI could choose an arm whose maximum optimistic value is lower and/or choose an arm having lower $x^T_{i,t}\mu$ but for which there are more possibilities for improvement."
> >
> > **Concerns about $\xi_{a(t), t}$ and the reward mean function $h$**
> >
> > We also fixed the concerns about $\xi_{a(t), t}$ and the reward mean function $h$. Please see Section 2 of our revised main paper.
> >
> > **Several combinations of statements are confusing**
> >
> > Based on your suggestions, we have now cleaned up these sentences. Thank you again for your valuable comments. Please let us know if you have any additional questions.
> >
> > **References**
> >
> > [4] K. Bhatia, K. Dahiya, H. Jain, P. Kar, A. Mittal, Y. Prabhu, and M. Varma. The extreme classification repository: Multi-label datasets and code, 2016.
> >
> > [5] Maxime Vono, Nicolas Dobigeon, and Pierre Chainais. High-dimensional gaussian sampling: A review and a unifying approach based on a stochastic proximal point algorithm. SIAM Review, 64(1):3–56, 2022.5

---

> > > ### Author Response · Authors · 2022-08-07
> > > **Followup on the previous discussion**
> > >
> > > Dear Reviewer,
> > >
> > > Now that the discussion period is coming to an end, we thank the reviewer for your constructive comments and suggestions!
> > > We also would like to explain more about your concern about the motivation of EI. Besides the competitive regret and competitive experimental results, the EI also has a particular benefit in practice in comparison with UCB/TS. Both UCB and TS strategies can be considered over-explored. In some situations, the exploration may costly or infeasible. For example, in medical decision-making, choosing a treatment that is not the estimated-best choice (pure exploration) for a specific patient may be unethical [1]. In this situation, our proposed EI chooses a safer treatment than UCB and TS. It is because EI only chooses a treatment with high possibilities for improvement over the estimated-best choice.
> > >
> > > Please let us know if you are unsatisfied with our responses or with our additional experiments during rebuttal.
> > >
> > > **References**
> > >
> > > [1] H. Bastani, M. Bayati, and K. Khosravi. Mostly exploration-free algorithms for contextual bandits. Management Science, 67(3):1329–1349, 2021.

---

> > > > ### Comment · Reviewer_s7GF · 2022-08-08
> > > > **LinEI vs LinTS and LinUCB**
> > > >
> > > > > We also would like to explain more about your concern about the motivation of EI.
> > > >
> > > > Scientifically I'm satisfied:
> > > >   * LinEI is statistically superior to LinTS; and
> > > >   * LinEI is computationally superior to LinUCB.
> > > >
> > > > Editorially, you need to state the above obviously.  The abstract has a sentence in it (good).  and L62 of the rebuttal version also helps.  But tables are really awesome.  If you have the space, consider a table with headings ("Technique", "Regret Bound", "Computationally Tractable?").

---

> > > > > ### Author Response · Authors · 2022-08-08
> > > > > **Many thanks for the feedback**
> > > > >
> > > > > Dear Reviewer,
> > > > >
> > > > > We are glad that our responses have addressed your main concerns regarding the performance of LinEI vs. LinTS and LinUCB. We just wonder if you would be kind to reflect your satisfaction with increasing your rating score?

---

> > > ### Comment · Reviewer_s7GF · 2022-08-08
> > > **Updated supplemental looks good**
> > >
> > > > LinEI scales better in high dimensions and in large action spaces (please see Figure 5 in Section A.2 of our updated Supplementary). This can be explained as the regret of LinEI is better than that of LinTS by a factor  while LinUCB is an over- exploration strategy.
> > > In terms of required computations, LinEI is significantly cheaper than LinUCB and LinTS (please see Table 3 in Section A.2 of our updated Supplementary).
> > >
> > > I see this, well done.  If space permits, provide a pointer to these results in the main text.

---

> > > > ### Author Response · Authors · 2022-08-08
> > > > **space + new version**
> > > >
> > > > Many thanks for your suggestions. We will definitely edit our paper to include these new results in the new version. We will also highlight the key strengths of LinEI and provide the requested pointers in this new version.

---

> > > > > ### Author Response · Authors · 2022-08-09
> > > > > **New version**
> > > > >
> > > > > Dear Reviewer,
> > > > >
> > > > > Based on your suggestions, we now edited the main text. To compare LinEI with LinUCB and LinTS, we added a table with headings ("Technique", "Regret Bound", "Computationally Tractable?") in the Introduction section. It looks good. We also added the additional experiment on the eurlex-4k dataset to the main text. Although the text space is exceeding 9 pages, we will seek to solve this issue in the next version. Please see our updated version in the Supplementary Material entry. Thanks for your suggestions!

---

> > > > > > ### Comment · Reviewer_s7GF · 2022-08-09
> > > > > > **wonderful**
> > > > > >
> > > > > > Table looks great.  Keep it in the main text.  The casual reader will see it first.
> > > > > >
> > > > > > Experiments, under space constraints, can be pushed to appendix with quick pointer in the main text.

---

> > > > > > > ### Author Response · Authors · 2022-08-09
> > > > > > > **Thanks**
> > > > > > >
> > > > > > > Thank you for your constructive comments on our paper. We will keep this table in the main text and will push several experiments to the appendix with a quick pointer in the main text. We are happy that you enjoyed our responses during the rebuttal.

---

### Official Review · Reviewer_rpMu · 2022-07-10

**Rating:** 6
**Confidence:** 3
**Soundness:** 3 good
**Presentation:** 3 good
**Contribution:** 3 good

**Summary:**

The authors propose a novel contextual bandit algorithm based on the expected improvement (EI) and study the corresponding regret analysis.  The proposed algorithm contains a modified element from EI for MAB by suggesting a hybrid of EI with pure exploitation. The paper adds different insights to the body of literature in that EI is an understudied technique to handle the tradeoff between exploration and exploitation in contextual bandits. They propose two novel EI-based algorithms for this problem, one for linear payoff and deep neural networks.  The authors provide numerical experiments.

**Questions:**

 Explanations for modifying EI by mixing with pure exploitation is not convincing.  If the EI value is small, that means the posterior variance is small, which will pick greedily anyway.  Is it necessary?  Also, what is the regret upper bound without the modification?

Do authors have experiments comparing TS and LinEI in terms of d?

**Limitations:**

yes

**Strengths And Weaknesses:**

Strength:

The paper adds different insights to the body of literature in that EI is an understudied technique to handle the tradeoff between exploration and exploitation in contextual bandits.

EI can be viewed as a variation of TS, and in that respect, the proposed algorithm improves the upper bound of TS by \sqrt{d}.

Weakness:

Explanations for modifying EI by mixing with pure exploitation are not convincing.

Why should one use the proposed algorithm over LinUCB since it carries extra \sqrt{log T}?

---

> ### Author Response · Authors · 2022-08-02
> **Please find responses to your questions below**
>
> We thank you for your encouraging and constructive feedback. We address your questions as follows:
>
> **Why should one use the proposed algorithm over LinUCB?**
>
> As the reviewer can see, our experimental results showed that LinEI outperforms LinUCB on various real datasets in practice. Especially in high dimensions, our additional experiments show that the proposed LinEI scales better and is computationally cheaper than LinUCB and LinTS as we will present below.  Moreover, UCB can be an over-explored strategy.  In some situations, the exploration may costly or infeasible. For example, in medical decision-making, choosing a treatment that is not the estimated-best choice (pure exploration) for a specific patient may be unethical. In these situations, our proposed hybrid of EI (an improvement over the estimated-best choice) with pure exploitation is more reasonable.
>
> **Explanations for modifying EI by mixing with pure exploitation is not convincing. If the EI value is small, that means the posterior variance is small, which will pick greedily anyway. Is it necessary? Also, what is the regret upper bound without the modification?**
>
> We thank you for your question. We now improve our explanation in the main paper. In fact, the modification comes from our theoretical view. If we use a pure EI (defined at Eq(1) of the main paper), we can reach a cumulative regret after $T$ iterations as
> $$R(T) \le \sum^T_{t=1} \beta_ts_{a(t),t} + \sum^T_{t=1} \sqrt{2\text{ln}(\frac{v_ts_{a(t),t}}{\alpha_{a(t),t}^{EI}})}v_ts_{a(t),t},$$ where $\beta_t, v_t$ are trade-off parameters; $s_{a(t),t}$ is the variance of reward of the arm $a(t)$ selected at iteration $t$;
> This is the best bound the authors can derive using existing concentration inequalities. While the first sum in the right side can be upper bounded by standard results in bandits, the second sum can go to infinity if $\alpha_{a(t),t}^{EI}$ goes to 0 at any round $t$. To avoid this issue, it is necessary for our regret analysis to take a threshold function into account as we explained in the main paper.  This allows us to bound the second sum sub-linearly in $T$. Moreover, when EI value is less than the threshold, we find that using a greedy strategy is better than the EI strategy in the sense that not only the regret is smaller (see our Lemma C.9 in Appendix) but also it is computationally cheaper. This is our interesting observation about the relationship between the EI and the greedy strategy.
>
> While our variant of EI in the linear setting can achieve a near-optimal regret, whether a pure EI converges is an open problem. We recall that for EI has been first designed for Bayesian optimization problems in the noise-free setting. Even in this setting, the current best result shows that the EI bears a sub-optimal regret bound $\mathcal O(T^{-\frac{1}{d}})$ [2], where $d$ is the input dimension and $T$ is the number of rounds. However, this upper bound is far from the lower bound of this problem $\Omega(T^{-\frac{\nu}{d}})$, where $\nu$ is the smoothness of the kernel used in the Gaussian process. The larger the $\nu$, the larger is the mismatch between the upper bound and lower bound. Thus, even for non-contextual problems simpler than contextual problems such as Bayesian optimization, whether the pure EI can achieve an optimal bound is still an open problem and was presented by [3] at the COLT 2022 conference. We have attempted to solve this problem by offering a modified EI variant (with near-optimal regret bound) while keeping it similar to the Pure EI.
>
> **References**
>
> [1] Shipra Agrawal and Navin Goyal. Thompson sampling for contextual bandits with linear payoffs. In Sanjoy Dasgupta and David McAllester, editors, Proceedings of the 30th International Conference on Machine Learning, volume 28 of Proceedings of Machine Learning Research, pages 127–135, Atlanta, Georgia, USA, 17–19 Jun 2013.
>
> [2] Adam D. Bull. Convergence rates of efficient global optimization algorithms. J. Mach. Learn. Res., 12:2879–2904, November 2011.
>
> [3] Vattar Vakili. Regret bounds for noise-free kernel-based bandits. 2020.

---

> > ### Author Response · Authors · 2022-08-02
> > **Please find responses to your questions below**
> >
> > **Do authors have experiments comparing TS and LinEI in terms of $d$?**
> >
> > Yes, we performed additional experiments during the rebuttal phase. In these experiments, we use contextual bandits with $K = 100$ actions. The context vectors are chosen uniformly at random from the unit ball. The reward function $r = x^T\theta^*$, where $\theta^*$ is generated \emph{a priori}. Using this setting, we compared our LinEI with LinUCB and LinTS in a range of dimensions $d =10, 100$ and $1000$.  We have provided a detailed description of our results in **our updated Supplementary material (please see our Section A.1**).
> > We summarise the results here. Our results show that
> >
> > 1. LinEI scales to high dimensions better than LinUCB and LinTS. This can be explained as the regret of LinEI is better than that of LinTS by a factor $\sqrt{d}$ while LinUCB can be over-explored in high dimensions (**please see Figure 4 in Supplementary file Section A.1**).
> >
> > 2. The computation time of LinEI is significantly smaller compared to that required by LinUCB and LinTS. LinUCB faces a NP-hard problem (as mentioned by [1] in their Related Work section) at each iteration, and thus spends more time for optimization, especially in high dimensions. LinTS uses a random sampling based heuristic and requires a Gaussian sampling step at each iteration. This is a high-dimensional Gaussian sampling problem (see [4]). It has computational costs and memory requirements that can rapidly become prohibitive in high dimensions. In contrast, our LinEI can avoid both the NP-hard problem and the high-dimensional Gaussian sampling problem. Thus, the computation cost of LinEI is significantly cheaper in high dimensions. (**please see Table 2 in Supplementary file Section A.1**).
> >
> > We also compare our proposed LinEI against LinUCB and LinTS on the eurlex-4k dataset which is a large-scale multi-label dataset in [5]. We have provided our results in **our updated Supplementary material (please see our Section A.2**).  It contains data with 5000 features and 3993 labels. It corresponds to a contextual bandit problem with 3993 actions and the dimension of context is $5000 + 3993 = 8993$ when mapped to our setting. This can be considered as a contextual bandit problem with large action space and with very high dimensions. Due to the heavy computation, we run algorithms on only 3000 datapoints from the test set. Our experimental results show that LinEI scales better to high dimensions and to large action spaces (**please see Figure 5 in Section A.2 of updated Supplementary**). In terms of required computations, LinEI is significantly cheaper than LinUCB and LinTS. The average-per iteration computation times of LinUCB, LinTS and LinEI are 1055.5s, 1496.2s and 181.3s respectively as shown in **Table 3 in Section A.2 of update Supplementary**. This is because LinUCB faces a very high-dimensional NP-hard problem at each iteration and LinTS faces a very high-dimensional multivariate Gaussian sampling problem as we explained above.
> >
> > **References**
> > [3]  Sartaj Sahni. Computationally related problems. 3(4), 1974.
> >
> > [4] Maxime Vono, Nicolas Dobigeon, and Pierre Chainais. High-dimensional gaussian sampling: A review and a unifying approach based on a stochastic proximal point algorithm. SIAM Review, 64(1):3–56, 2022.
> >
> > [5] K. Bhatia, K. Dahiya, H. Jain, P. Kar, A. Mittal, Y. Prabhu, and M. Varma. The extreme classification repository: Multi-label datasets and code, 2016.

---

### Official Review · Reviewer_XWbh · 2022-07-12

**Rating:** 8
**Confidence:** 3
**Soundness:** 3 good
**Presentation:** 3 good
**Contribution:** 4 excellent

**Summary:**

The paper shows how to extend the widely-used expected improvement heuristic from into the contextual bandits setting to create a new basic type of contextual bandit algorithm. They propose two novel algorithms, and propose a method for choosing an improvement threshold for controlling the exploration/exploitation cutoff which provably achieves an $\tilde O(\sqrt{T})$ regret rate, even in settings with adaptive adversaries. The paper then shows that the proposed methods have strong empirical performance.

**Questions:**

1) Are $\Phi$ and $\phi$ supposed to be the cdf and pdf of the normal distribution? They seem to be introduced on line 143 without that being explicitly stated.

**Limitations:**

The paper discusses the limitations of using a Gaussian distribution for the reward model, as well the fact that their usage of the NTK kernel restricts the neural network classes that they can use.

The paper does not address potential negative societal impact of this work.

**Strengths And Weaknesses:**

Strengths:
1) Paper extends the expected improvement heuristic into contextual bandits, building a connection with the best arm identification and bayesian optimization literature, providing a novel and significant result.
2) Paper provides proofs for the results, and is able to get competitive regret rates for linear contextual bandits and neural bandits.
3) Paper shows experimental evidence for the value of the method.
4) Paper is clearly written, and contextualizes expected improvement in the broader literature, and clearly shows how the analysis of the linear and neural cases differ.

Weaknesses:
1) (minor) I wish the paper explained a bit more how the algorithms are able to work even with an adaptive adversary.

---

> ### Author Response · Authors · 2022-08-02
> **Please find responses to your questions below.**
>
> We thank you for your encouraging and positive feedback on our paper. We hope that EI will become a promising approach compared to existing approaches in contextual bandits and further in reinforcement learning.
>
> Regarding the question:
>
> **I wish the paper explained a bit more how the algorithms are able to work even with an adaptive adversary.**
>
> We use an adaptive adversary in the sense that all context vectors can be arbitrarily chosen. Our proposed LinEI works in this regime because all concentration bounds we have used for the regret analysis hold without constraints on context vectors. LinTS also works with an adaptive adversary, however, the regret bound of our novel LinEI improves that of LinTS by a factor $\sqrt{d}$, where $d$ is the dimension of context.
>
> Very recently, a paper at NeurIPS 2021 [1] improved this regret bound by integrating a doubly robust estimator with TS. They obtain a regret bound $\tilde{\mathcal O} (\phi^{-2}\sqrt{T})$ where $\phi^2$ is the minimum eigenvalue of the covariance matrix of contexts. they show that $\phi^{-2} \ge d$ in general and $\phi^{-2} = d$ in only some cases. Thus, this regret bound is not near-optimal in general. In addition, this work requires additional significant computations and their setting is limited to independent contexts and requires the minimum eigenvalue of the covariance matrix of contexts to be bounded. In contrast, our proposed LinEI obtains a near-optimal regret in a general setting where we do not require the above conditions on contexts. This shows the benefit of EI for regret analysis compared to the TS-based approach.
>
>
> **Are $\Phi$ and $\phi$  supposed to be the cdf and pdf of the normal distribution? They seem to be introduced on line 143 without that being explicitly stated.**
>
> Yes, you are right. They are the cdf and pdf of the normal distribution.
>
> References
>
> [1] Wonyoung Kim, Gi soo Kim, and Myunghee Cho Paik. Doubly robust thompson sampling for linear payoffs, 2021.

---

### Meta-Review · Area_Chair_5p93 · 2022-08-26

**Recommendation:** Accept
**Confidence:** Certain

**Metareview:**

This paper proposes and analyzes algorithms based on expected improvement for the contextual bandit setting, and proves that the resulting algorithm can attain $O(d \sqrt{T})$ regret in the linear bandit setting (the result improves over Linear TS).

All the reviewers agree that the modified LinEI algorithm and its analysis are novel and important to the community. I agree with the reviewers, and recommend accepting the paper. For the final version of the paper,  it would be helpful to add more details on why the pure EI strategy does not work and add the scaling with $d$ experiments to the main paper (the response to Rev. rpMu). If there is space, it would also be helpful to add a proof sketch for the LinEI algorithm and distinguish it from the LinTS analysis which is more standard and known in the community (response to Rev. KPAe).

**Award:**

No

---

### Decision · Program_Chairs · 2022-09-14

Accept